# Single-breath-hold 3D abdominal metabolic MRI enables label-free diagnosis of liver cancer

Chuyu Liu[1,5], Nan Gao[1,5], Haiqi Ren[2], Hao Liu[2], Juxiang Hou[1], Zhongsen Li[1], Benqi Zhao[3], Yibei Yu[3], Xiaowei He[2], Zhuozhao Zheng[3] & Xiaolei Song [1,4] ✉

Chemical exchange saturation transfer (CEST) MRI could detect proteins/peptides, creatine, glucose, and glycogen by labeling their exchangeable amide, amine, and hydroxyl groups respectively, via frequency-selective RF pulses. Without the need for contrast agents or specialized hardware, CEST can be conveniently integrated into existing clinical MR protocols. However, its abdominal application is limited by long scan time (> 5 min) and susceptibility to respiratory motion (60–70% successful scan rate). We develop an ultra-fast 3D CEST MRI approach using spatial-spectral encoding (SSE), enabling a full spectral scan of whole-liver 3D images within a single breath-hold. SSE-CEST employs an efficient $z$-$\omega$ encoding pattern by applying a saturation gradient, followed by a data-driven spatial spectral reconstruction based on the low-rankness of CEST spectra. SSE-CEST is comprehensively evaluated in glycogen phantoms, ex vivo porcine liver, healthy volunteers and patients. Single breath-hold SSE-CEST largely improves successful rate, with a correlation of 0.95 between two repeated scans. SSE-CEST enables the detection of multi-metabolite changes in the liver and pancreas after an overnight fasting, and the dynamic mapping of hepatic glucose metabolism during an oral glucose test. For liver cancer patients, SSE could differentiate active lesions from post-treatment necrosis, featuring superior in-slice spatial resolution and motion-stabilized images. SSE-CEST MRI potentially could facilitate the diagnosis and patient management for liver and other abdominal diseases.

The liver, the largest gland in the human body, plays vital metabolic roles, including food digestion, glycogen storage, and the synthesis and secretion of proteins, fats, and vitamins. Altered liver metabolism is linked to various chronic and acute diseases, such as cancers, diabetes, glycogen storage diseases, and other metabolic disorders, affecting one third of the global population[1].

Magnetic resonance imaging (MRI) is a powerful non-invasive tool for managing liver disease patients, allowing precise localization and functional evaluation of both focal and diffuse lesions[2]. Additionally, MR spectroscopic methods, including 1H-MRS[3], 13C-MRS[4,5] and 31P-MRS[6], have shown promise in revealing liver metabolic functions including fat characterization, glycogen and glucose metabolism, and

[1]Center for Biomedical Imaging Research, School of Biomedical Engineering, Tsinghua University, Beijing, China. [2]School of Information Sciences and Technology, Northwest University, Xi'an, Shaanxi, China. [3]Department of Radiology, Beijing Tsinghua Changgung Hospital, Beijing, China. [4]Present address: School of Medical Technology, Beijing Institute of Technology, Beijing, China. [5]These authors contributed equally: Chuyu Liu, Nan Gao. ✉e-mail: songxl@tsinghua.edu.cn; xlsong8@163.com

mitochondrial functions. However, these MRS methods either have low sensitivity or require specialized agents and hardware, which hinders their clinical application. Recently, a label-free metabolic MRI technique, called chemical exchange saturation transfer (CEST), has demonstrated the capability of detecting tissue endogenous glycogen[7,8], protein content[9] and creatines[10]. Besides, dynamic MRI of glucose metabolism can be obtained following oral or intravenous glucose uptake[11].

As a convenient add-on to routine MR protocols, CEST has demonstrated substantial values in the diagnosis and prognosis of glioma patients[12,13], and has become an FDA-approved product. These values include tumor differentiation and grading[14], identifying gene mutations[15], and monitoring treatment response[16]. However, the application of CEST in abdominal imaging has been significantly limited, due to the prolonged scan time and susceptibility to respiratory motion[17]. Only two clinical studies have been published, displaying low scan success rates (~60–70%) and inconsistencies among observers and studies[18–20].

As a spectral-based method, CEST-MRI requires multiple well-aligned images at a series of saturation frequencies ($\omega$), for field correction and signal quantification. CEST acquisition for each $\omega$ typically takes 4–6 s, involving second-long saturation RF pulse for labeling certain types of protons and transferring their signals to water for amplification, followed by image readout at water frequency. To acquire CEST images from both amide (mobile proteins and peptides) and hydroxyl (glycans) groups[21], tens of saturation $\omega$'s are required,

leading to minute-long scan time. Therefore, the technique is not reliable for abdominal exams, easily corrupted by image misalignment or by readout artifacts. By using respiratory triggers[22,23] or data-inherent rhythms[24], free-breathing liver CEST techniques have been developed and tested on healthy subjects. But their clinical applications are still hindered by non-neglectable image misalignment among respiratory cycles, as well as the severely-prolonged scan time (> 5 min)[18,19].

Ideally, a single-breath-hold 3D abdominal CEST sequence is preferred for both scan efficiency in clinical settings and improved image quantification. Herein, we propose a spatial-spectral encoding (SSE) metabolic MRI technique that enables super-fast abdominal CEST within a single breath-hold. The SSE method is inspired by ultra-fast CEST-spectral (UFZ) acquisition, which simultaneously applies a gradient field with the saturation RF pulse. UFZ enables fast scanning of a tube[25], a selected voxel[26], or a 10 mm-thick slab[27], but it fails to resolve 3D spatial distribution as effectively as conventional multi-slice or 3D sequences.

When expanding each slice of CEST dataset along the voxel dimension, a $z$-$\omega$ plane formed with a single UFZ scan corresponding to a diagonal line (Fig. 1). To efficiently sample the $z$-$\omega$ plane and thereby resolve spatial-spectral information, we designed a 'diagonal' acquisition trajectory composed of multiple UFZ scans. Next, based on the low-rankness of CEST spectra, voxels within a slice or adjacent slices were grouped together for spectral feature extraction, achieving

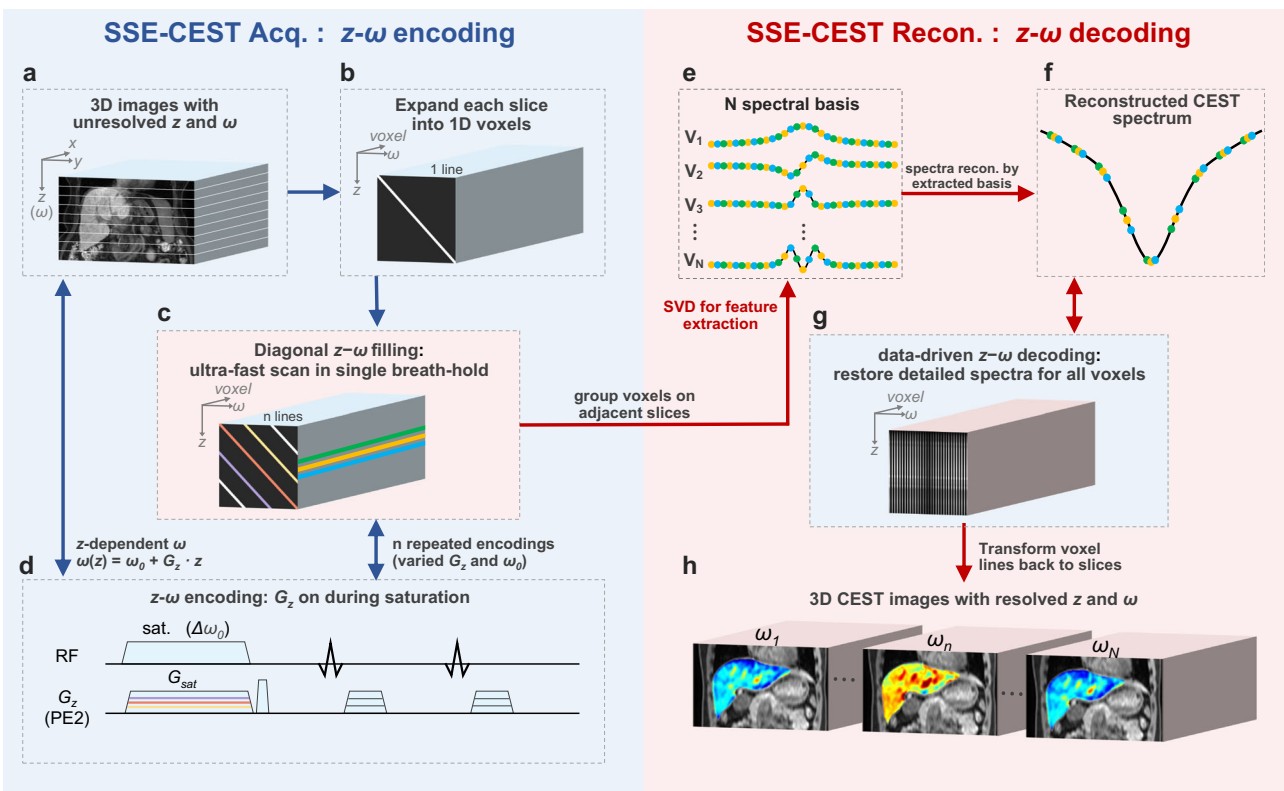

**Fig. 1 | Single breath-hold metabolic MRI framework via spatial-spectral encoding (SSE) CEST.** SSE-CEST is achieved by efficient $z$-$\omega$ encoding during acquisition (**a**–**d**) and accurate $z$-$\omega$ decoding during reconstruction (**e**–**h**). Briefly, 3D liver images (**a**) were acquired using a $z$-$\omega$ encoded MRI sequence (**d**) by applying a gradient along slice-direction during saturation ($G_z$ here, also termed as $G_{sat}$). One $z$-$\omega$ encoding step is a diagonal line in the $z$-$\omega$ plane when expanding voxels for each slice along voxel direction (**b**). To efficiently fill the $z$-$\omega$ plane, multiple diagonal $z$-$\omega$ lines were acquired via a designated pattern (**c**), consisted of multiple $z$-$\omega$ encoding steps with varied $G_{sat}$ and $\omega$ (**d**). In the acquired raw dataset (**c**), CEST spectra were loosely-sampled in $\omega$ (1–2 ppm intervals) with distinct

frequency offsets for each slice. To restore the detailed CEST spectra, voxels within every three adjacent slices are fused to extract their spectral features, i.e., spectral basis function calculated by Singular Value Decomposition (SVD) (**e**). By fitting the densely-sampled spectral bases to the acquired frequency offsets, a CEST spectrum with small intervals could be interpolated from the basis functions (**f**, **g**), and the entire dataset with resolved $z$ and $\omega$ could be obtained (**h**). After transforming each voxel line back to $x$-$y$ slice, a resolved 3D imageset could be obtained for each $\omega$, resulting in reconstructed data containing ~5000 saturation images (41 slices × 100–200 $\omega$).

customized spectral reconstruction of each slice. The efficient acquisition strategy of multiple $z$-$\omega$ encodings, combined with data-driven CEST-spectral interpolation for each slice, makes SSE a revolutionary technique for CEST metabolic MRI in clinical settings. Within a single breath-hold, SSE acquires high-quality 3D images at multiple frequency offsets targeting amides, amines, and hydroxyls, and also allows fine quantitative images, including APTw, GlucoCEST, and Lorentzian-fitted images. Furthermore, the high in-slice resolution allows recognition of small lesions in the liver or kidneys, as well as small organs such as the pancreas and spleen.

## Results

### Imaging framework

Super-fast 3D spatial-spectral encoding CEST (SSE-CEST) MRI is accomplished through two steps: the $z$-$\omega$ encoding step during acquisition and the $z$-$\omega$ decoding step during reconstruction (Fig. 1). To rapidly acquire multiple saturation frequencies, we adopted a $z$-$\omega$ encoding sequence, which applies a field gradient along the $z$-dimension ($G_{sat}$) during the saturation RF pulse. This results in a $z$-dependent saturation frequency ($\omega$) for different slices, corresponding to a diagonal line in the $z$-$\omega$ plane (Fig. 1a, b).

$$\omega(z) = \gamma G_{sat} \cdot z + \omega_0 \qquad (1)$$

By varying either the center frequency $\omega_O$ or $G_{sat}$, efficient traversal and sampling of the $z$-$\omega$ plane is achieved through multiple $z$-$\omega$ encodings arranged in a designated pattern (e.g., parallel, radial, diamond) (Fig. 1c). In the reconstruction step, we grouped voxels within each slice and adjacent slices to extract spectral features via Singular Value Decomposition (SVD). Due to the low-rank nature of CEST spectra, a slice-specific interpolation is performed using these features, i.e., spectral basis functions (Fig. 1d, e, Supplementary Fig. 1.1). Noted that SSE-CEST reconstruction utilized the same Partial Separable theory as in the MRSI accelerations[28], the spatial resolution does not be affected, even when grouping voxels on adjacent slices for augmentation of sampled $\omega$ (each slice presents loosely-sampled and distinct $\omega$ (Supplementary method 1). Without any truncation, the image sets can be accurately resolved by poly-nominal fitting the spectral bases and transforming the voxel dimension back to 2D slices (Fig. 1f, g). For human abdominal protocol, we acquire only 10 or 11 $z$-$\omega$ lines plus $S_0$ within a single breath-hold (18–21 s), enabling 3D CEST images with a fine-spectral of 121 saturation frequency offsets (>11× spectral interpolation).

### Ex-vivo porcine liver: flexibility to $z$-$\omega$ acquisition patterns

SSE-CEST was first validated on an ex-vivo scan of porcine liver, with three distinct $z$-$\omega$ acquisition patterns tested: parallel, diamond, and radial (Fig. 2a1-a3). The three patterns under a B1 of 0.7 µT are as follows: parallel (4 min 14 s, 30 $z$-$\omega$ lines); diamond (4 min 22 s, 31 $z$-$\omega$ lines); radial (2 min 38 s, 18 $z$-$\omega$ lines). In contrast, the conventional CEST acquisition without $z$-$\omega$ encoding took 11 min 58 s with 88 $\Delta\omega$ values (Fig. 2a4). Three types of contrast maps were obtained by a Lorentzian Difference (LD) analysis (Supplementary methods, Supplementary Table 1), including amide CEST, glycogen NOE and aliphatic NOE (Fig. 2b1-b4). At the bottom of each SSE-CEST contrast map, Bland-Altman analysis revealed its consistency with that acquired using the conventional method, with all three patterns exhibiting acceptable 95% limits of agreement (parallel ≤ [−2.58%, 1.37%], diamond ≤ [−2.36%, 1.15%], radial ≤ [−2.63%, 2.14%] as quantitatively listed in Supplementary Table 1.2, $n = 69$, the number of liver ROIs ($8 \times 8$ square) within the displayed slice). When compared with the conventional methods, none of the SSE-CEST contrast maps displayed statistically significant biases ($p > 0.05$, $n = 69$, same as above) except for aliphatic NOE with radial pattern (Supplementary Fig. 2.1).

Spectral consistency was further validated through averaged Z spectra and the substitute LD spectra within a ROI (Fig. 2d), where parallel and diamond patterns achieved exceptional spectral reproducibility, outperforming radial pattern (Z-spectra: parallel RMSE = 0.69%, diamond RMSE = 0.67%, radial RMSE = 1.94%; LD spectra: parallel RMSE = 0.59%, diamond RMSE = 0.54%, radial RMSE = 1.47%, Fig. 2c1-c3). SSE-CEST demonstrated well spatial reconstruction with vessel details for all acquired slices, as shown on the raw ST images at 3.5 ppm, −1.2 ppm and 1.2 ppm, respectively. (Supplementary Fig. 2.2) Bland-Altman analysis also suggested high agreement with the conventional method and minimal variance across slices (Supplementary Fig. 2.3; $n = 21$ slices, one ROI per slice at the location indicated in Fig. 2d).

### Quantitative evaluations using glycogen phantoms

Glycogen phantoms with various concentrations, dissolved in 1% Agar, were scanned for quantitative assessment. As CEST spectral profile changes with saturation $B_1$, we acquired a lower $B_1$ for measuring the relayed-NOE from macromolecular glycogen (0.7 µT, Fig. 3) and a higher one for enhancing the hydroxyl CEST signals on glycans (2 µT, Supplementary Fig. 3.1).

The glycogen-NOE maps derived from 0.7 µT SSE-CEST show consistency among slices (represented slices. #9, 11, 13, Fig. 3b, c), revealing linear correlation with the glycogen concentrations (Fig. 3d, $n = 21$ slices, mean ± std, r = 0.95). Averaged Z-spectra per tube in slice 11 (Fig. 3e) illustrated concentration-dependent profiles, with the marked points indicating ×8 under-sampling of SSE-CEST. Although distinct saturation frequencies were sampled on each slice, the reconstructed Z-spectra of SSE-CEST were almost identical for the same tube (Supplementary Fig. 3.2). LD spectra extracted from raw Z-spectra further displayed cleaner glycogen-NOE spectral peaks (−0.5 to −2 ppm) and hydroxyl signals (0.5 to 2 ppm) on a quantitative manner (Fig. 3g).

SSE-CEST demonstrated high-fidelity reconstruction when compared with full-sampled conventional CEST, as suggested by the scattering plots and the Bland-Altman plots in Fig. 3g, h (0.7 µT), Supplementary Fig. 3.1 (2 µT) (all with $n = 126$, 6 tubes X 21 slices, glycogen NOE signals: r = 0.97, $p < 0.0001$, mean difference 0.25%, 2 µT ST(1.2 ppm) and ST(−1.2 ppm), both with $r = 0.98$, $p < 0.0001$), and Supplementary Fig. 3.5. Slice-by-slice comparison of SSE-CEST and conventional CEST ST images demonstrated spatial consistency and correlation with glycogen concentration. (Supplementary Fig. 3.3, 3.4).

### Single-breath-hold 3D abdominal CEST MRI in healthy volunteers

On ten healthy volunteers, we demonstrated a parallel SSE-CEST MR protocol within a single breath-hold (18-21 seconds, $B_1 = 2$ µT). By collecting 11 parallel $z$-$\omega$ lines, SSE-CEST enabled reconstruction of 41-slice saturation images across 121 frequencies, achieving 44-fold (11-fold spectral interpolation multiplied by 2×2-acceleration upon parallel imaging) acceleration compared with conventional fully-sampled approach (Fig. 4a). By acquiring an additional reference without saturation ($S_0$), SSE-CEST could directly output the ST maps (ST = 1-S($\Delta\omega$)/$S_0$), reflecting contrasts from amides from mobile proteins/peptides (3.5 ppm), hydroxyls on glycans (1.2 ppm),and the relayed NOE from macromolecular aliphatic protons (−3.5 ppm)[8,29]. Noted that SSE-CEST utilized a Dixon-type readout with capability of water-fat separation and B0 maps, readily providing B0-corrected images without fat contaminations. The B0-corrected APTw images as used in clinical glioma diagnosis were also obtained (MTR$_{asym}$@3.5 ppm, Fig. 4b, c, f)[13]. Consecutive scan-rescan data showed high repeatability of SSE-CEST, with 24 out of the 40 contrast images exhibiting a correlation of > 0.99 between the two scans and only one contrast map showing low correlation (Fig. 4b). All 40 evaluated contrast maps displayed low mean squared error between two repeats (Fig. 4c

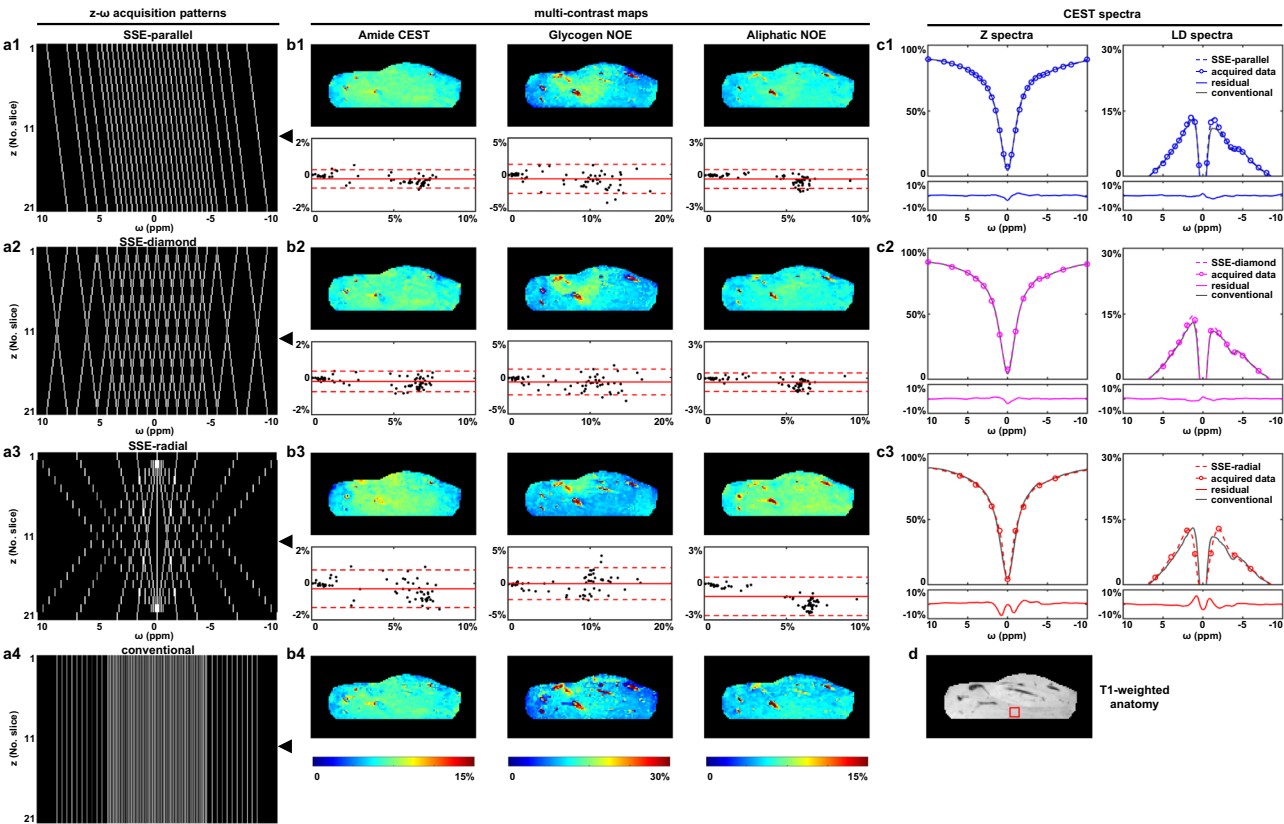

**Fig. 2 | Ex vivo porcine liver metabolic imaging with SSE-CEST using variable z-ω acquisition patterns.** Parallel (**a1**–**c1**), diamond (**a2**–**c2**), and radial (**a3**–**c3**) SSE-CEST are compared with conventional CEST (**a4**–**b4**). The left column (**a1**–**a4**) shows the respective acquisition patterns in z-ω plane. The middle columns (**b1**–**b4**) present multi-contrast maps (Amide CEST, Glycogen NOE, and Aliphatic NOE) from a representative axial slice, with their locations in z-ω plane marked by black arrows in (**a1**–**a4**). Bland-Altman analysis (the bottom plots in (**b1**–**b3**), $n = 69$, the number of liver ROIs (8×8 square) within the displayed slice) demonstrates good agreement between each SSE-CEST pattern and conventional CEST. Mean difference in solid lines and 95% Limits of Agreement in dashed lines (see quantitative results in Supplementary Table 1.2). The right columns (**c1**–**c3**) display CEST spectra (Z and LD spectra) from the region of interest indicated by a red open box in the T1-weighted anatomy (**d**). Spectra are color-coded (SSE-parallel: blue, SSE-diamond: magenta, SSE-radial: red; dashed lines for SSE-CEST, solid gray lines for conventional CEST). The near-zero difference spectra (solid lines around zero) confirm minimal deviation between SSE-CEST and conventional CEST. All data were acquired with B1 = 0.7 µT and quantified using LD. Further statistics and comparison across all slices are in Supplementary Table 1.2 and Supplementary Figs. 2.1–2.3.

APTw< 0.1% for all subjects). Pearson correlation and the Bland-Altman analysis from average liver signal from all ten subjects further suggested the reliability of SSE-CEST. (Supplementary Fig. 4.2, $r = 0.95$ for ST@1.2 ppm, $r = 0.96$ for ST@3.5 ppm). For one of the subjects, the multi-metabolite ST images, the APTw images and B0 maps of the entire liver are displayed in three orthogonal slices, overlaid on a high-resolution T1w images (Fig. 4d, e). The single-breath-hold 3D SSE-CEST protocol provides excellent repeatability and capability of multi-modality fusion, towards a robust metabolic MRI tool in clinic.

### Overnight fasting: multi-metabolite, multi-organ MRI
The SSE-CEST MRI was evaluated in a fasting experiment involving five healthy volunteers. Each subject underwent an initial scan at 2 h after a full dinner and was re-scanned after a 12 h fast. Each scan involved three single-breath-hold SSE protocols, which are Zpos (0.7 µT), Zneg (0.7 µT) and Z (2 µT), each with 11 parallel z-ω lines plus $S_0$ (Supplementary Table 1, Fig. 5.1, same protocol in tumor patients). For SSE-CEST using a saturation B1 of 0.7 µT, three types of contrast were calculated, including Amide CEST, Glycogen NOE and Aliphatic NOE. Whereas for SSE-CEST of 2 µT, the APTw maps and the glycan-CEST maps were obtained using asymmetric MTR analysis. Figure 5a, b displayed the five contrast maps from one of the subjects, revealing notable signal drops post overnight fasting in amide CEST and glycogen NOE (~1/3), as also denoted in the LD spectra of a liver ROI

(Fig. 5c). When comparing all 205 slices acquired from 5 subjects acquired post-meal and post-fasting, all five contrast maps displayed significant difference (Fig. 5d). To further eliminate the changes in water relaxation, another R1ρ-based quantification method was also performed, validating the signal drops of glycogen NOE and of amide CEST (Supplementary Fig. 5.5). The multi-offset ST images also displayed significant signal drops post fasting, with the high-quality motion-free images allowing for easy recognition of major vessels and the multiple organs including spleen, pancreas, kidney. (Supplementary Figs. 5.6, 5.7). Noted the 2 µT SSE-CEST images may present lower SNR than the 0.7 µT ones when offsets getting closer to water, suggesting the benefit of using a smaller B1, with lower contaminations from non-CEST background signals.

### Dynamic Glucose Imaging for Oral Glucose Tolerance Test (OGTT)
To demonstrate the utility of SSE-CEST for dynamic spatial-spectral characterization of biological systems after administration of exogenous agents, we conducted an OGTT Experiment in five healthy human volunteers[11,30–32]. Each volunteer underwent repeat SSE-CEST scans until 51 minutes after glucose uptake, and a pre-administration baseline scan was also acquired (Fig. 6a). SSE-CEST present a unique advantage of full-spectral capture, allowing for monitoring the spectral profile over time and making the CEST data more interpretable and

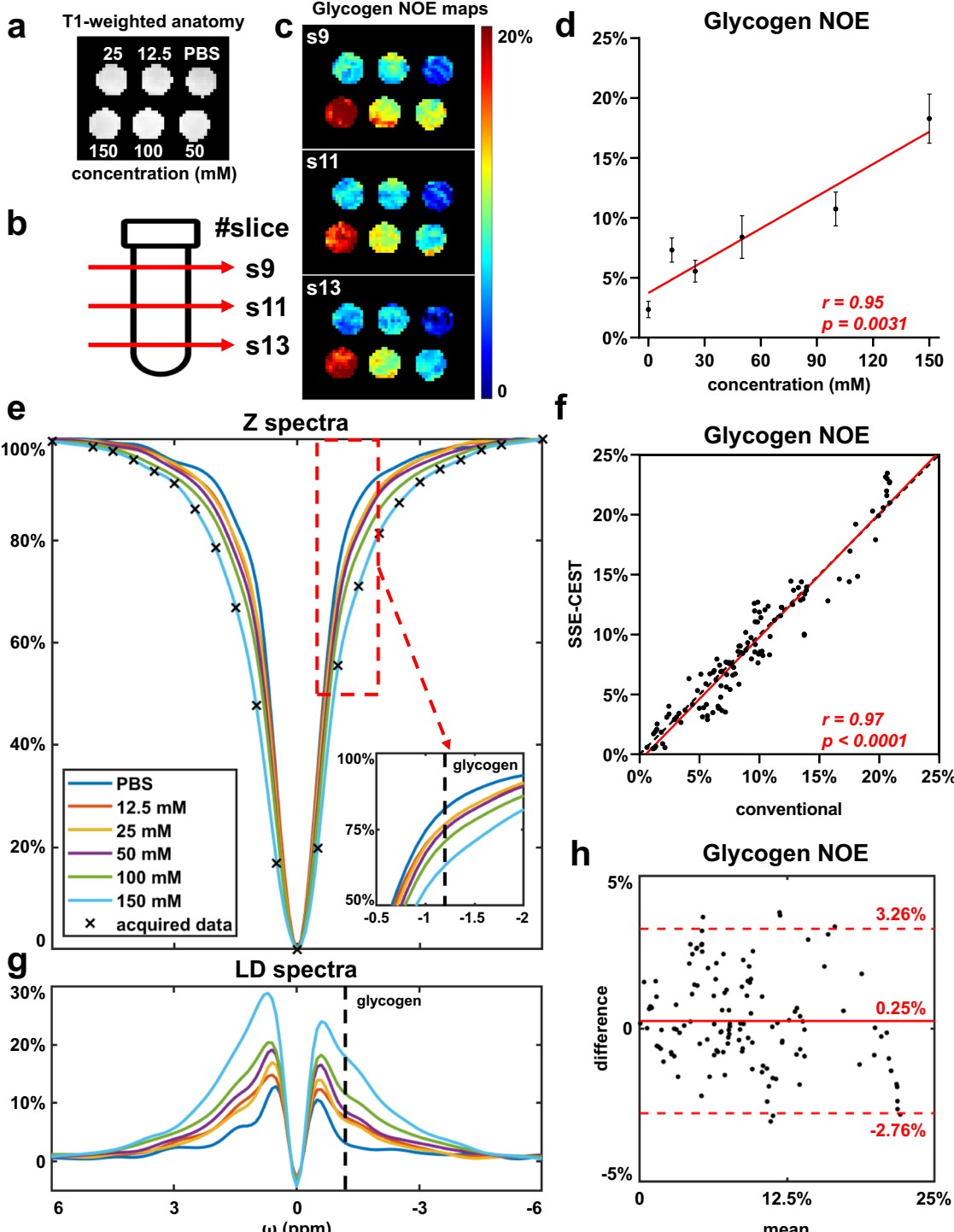

**Fig. 3 | Quantitative evaluations of SSE-CEST using glycogen phantoms in 1% agar. a** Phantom layout in a *x-y* slice. **b** Illustration of different slices along *z* direction. **c** Glycogen NOE (LD at −1.2 ppm) maps for slice #9, 11, 13, displaying a linear-correlation with the glycogen concentration (**d**, two-sided Pearson correlation, *p* = 0.0031, *n* = 21 slices, mean ± std.) **e** Z spectra from SSE-CEST for each tube in slice #11, with the symbol 'x' marked the actual acquisition points. The specific frequency range from glycogen NOE are zoomed in the right bottom (−0.5 ppm to −2 ppm). **f, h** Scattering plots and Bland-Altman plots of glycogen NOE values,

comparing between SSE-CEST and conventional CEST (both with *n* = 126, 6 tubes × 21 slices; (**f**), two-sided Pearson correlation, *p* < 0.0001). **g** CEST LD spectra derived from (**d**), highlighting cleaner NOE signal peaks (−0.5 to −2 ppm) and hydroxyl signals (0.5 to 2 ppm). B1 = 0.7 µT and parallel *z-ω* pattern were used for this figure. Results for other acquisition parameters are displayed in Supplement Figs. 3.1 and 3.3–3.4, including slice-by-slice image comparison and statistical comparison with fully-sampled conventional CEST, and quantitative correlation with glycogen concentration.

reliable. Figure 6b compared Z-spectra and derived MTR$_{asym}$ spectra acquired the baselines and at 51 min post take-up, with the largest changes observed at -2.1 ppm similar to previous findings in brain[32]. For a representative subject, Fig. 6c presents dynamic glucose images

quantified via MTR$_{asym}$ (2.1 ppm) and the corresponding time course of signals within liver voxels; signals rose consistently until ~30 min after oral glucose administration and were maintained through the final scan (51 min). Whereas the upward trend of T2 values is less obvious,

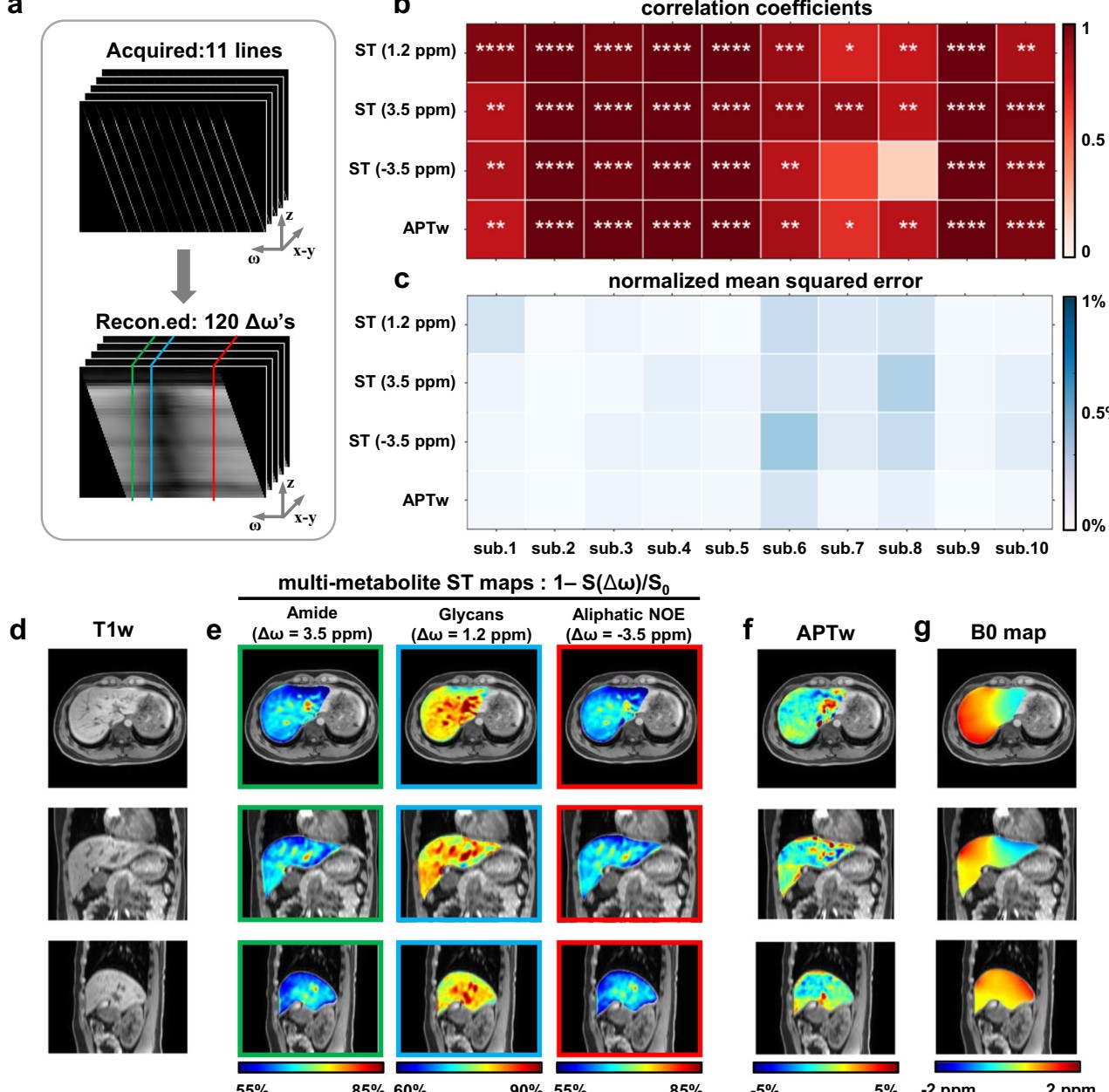

**Fig. 4 | SSE-CEST for metabolic imaging in healthy volunteers. a** Illustration of the acquisition pattern (11 $z$-$\omega$ lines plus a reference S0) and the reconstructed fine-interpolated $z$-$\omega$ space (−6 ppm to 6 ppm with 0.1 ppm intervals for the center slice), all with single-shot 3D DIXON-TFE readouts under B1 = 2 μT (Supplementary Fig. 4.1). **b, c** Repeatability of SSE-CEST evaluated on ten healthy subjects (sub.1-sub.10), correlation coefficients are in red and normalized mean squared error are in blue. (**b**, two-sided Pearson correlation, $n$ = 10 slices per subject, correlation coefficients and $p$ values are shown in Supplementary Table 1.3; **c** detailed data are shown in Supplementary Table 1.4). **d** anatomical images. **e** multi-metabolite CEST images of amide (3.5 ppm), glycans (1.2 ppm), and aliphatic NOE (−3.5 ppm). **f** APTw images obtained by the subtraction of previous images at 3.5 ppm and −3.5 ppm. **g** B0 maps obtained by the Dixon readouts. *, **, ***, **** denote $p < 0.05$, $p < 0.01$, $p < 0.001$, $p < 0.0001$, respectively.

only displaying <6 ms (mean T2 = 101.9 ms) changes of mean signals with a larger standard deviation, presumably due to less specificity and the single-slice acquisition we used. For all five subjects, liver MTR$_{asym}$ maps showed consistent signal increases within 15 min post-administration, with low inter-voxel variability across parenchymal voxels; (Supplementary Fig. 6.1). The time course also varied among subjects, possibly due to differences in individual digestion times after oral administration. Nevertheless, images acquired at all time-points displayed well-alignment, clearly depicting the pancreas. These evidences suggested SSE-CEST allows for rapid capture of the dynamics post

administration of exogenous agents, with good spatial-spectral performance.

## Assessment in patients with focal lesions

We further evaluated SSE-CEST in patient with Hepatocellular Carcinoma (HCC). Two types of lesions were included in this study: active HCC lesions and necrotic lesions after treatment. Figure 7a, b show representative images from a patient who presents both active HCC and necrotic lesions 4 months post-treatment, as characterized on routine clinical contrast-enhanced images. Three SSE-CEST scans were

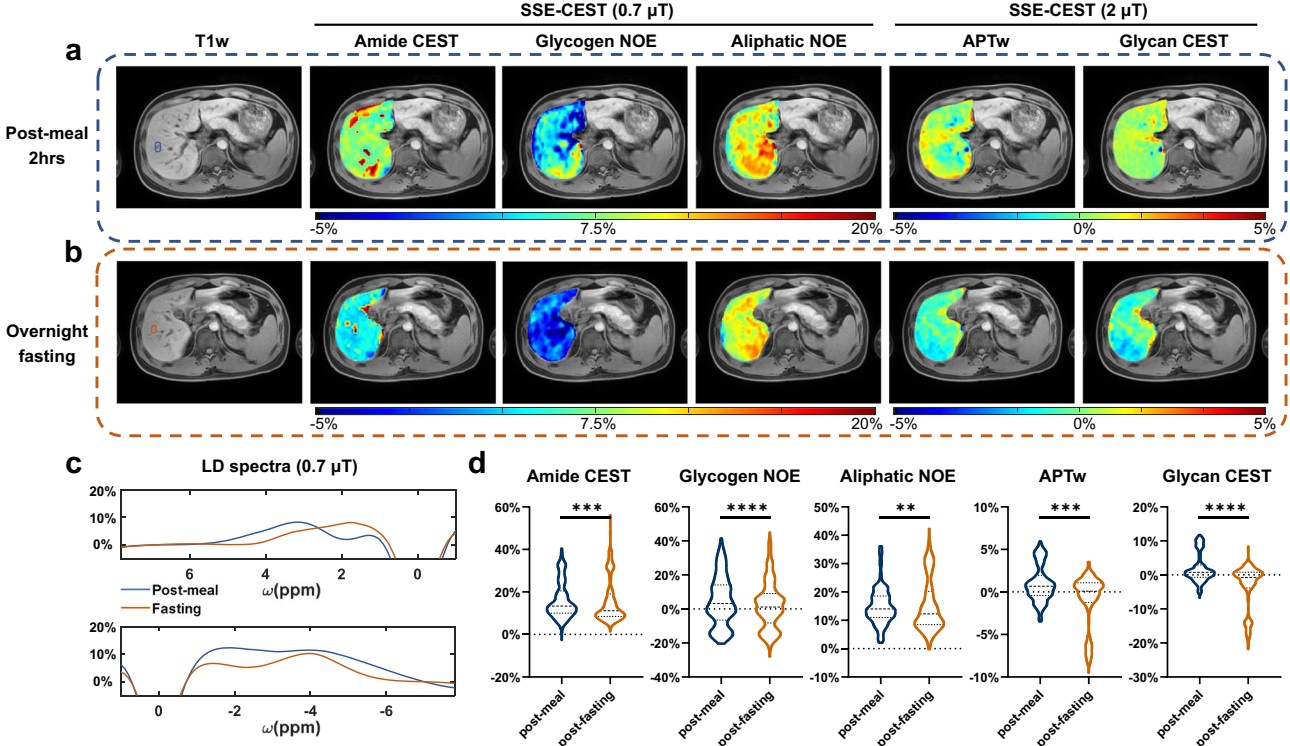

**Fig. 5 | SSE-CEST for evaluating metabolic changes in the liver during fasting experiments.** For a representative healthy volunteer, (**a**) the anatomical and metabolic images (from left to right including Amide CEST (Zpos-0.7 µT), Glycogen NOE (Zneg-0.7 µT), aliphatic NOE (Zneg-0.7 µT), APTw (Z-2 µT) and Glycan CEST (Z-2 µT) acquired two hours after the meal, (**b**), same layouts as **a**, but acquired after overnight fasting. **c** Compared the LD spectra at two time points (blue: Post-meal; orange: overnight-Fasting), for a ROI denoted in (**b**). Upper row is from the Zpos protocol and the bottom row is from Zneg protocol. The LD analysis details were displayed in Supplementary Figs. 5.2-5.4. **d** The statistical comparison of five contrast maps acquired within 2-hr post-meal and post overnight fasting (two-sided paired *t*-test, $n = 205$ slices, with 41 slices from each subject; Amide CEST $p = 0.0003$, Glycogen NOE $p < 0.0001$, Aliphatic NOE $p = 0.0083$, APTw $p = 0.0006$, Glycan CEST $p < 0.0001$). The Rex images and spectra of the same slice were displayed in Supplementary Fig. 5.5. The ST images and Z-spectra of multiple abdominal organs are shown in Supplementary Figs. 5.6–5.7. *, **, ***, **** denote $p < 0.05$, $p < 0.01$, $p < 0.001$, $p < 0.0001$, respectively.

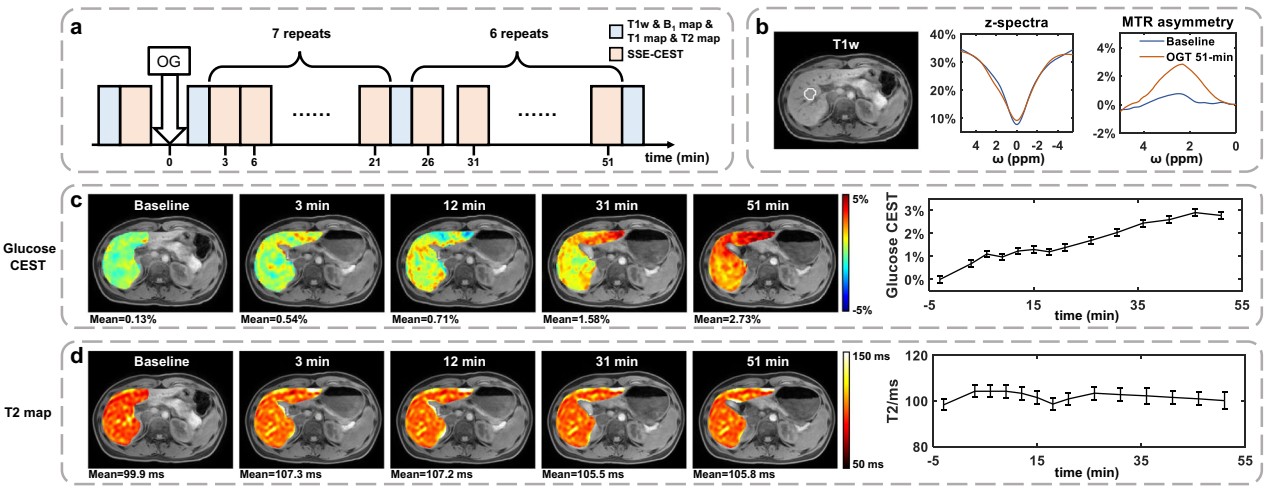

**Fig. 6 | SSE-CEST for dynamic glucose imaging during oral glucose tolerance test (OGTT) experiments. a** MRI scan protocol of OGTT experiments, which consists of 13 SSE-CEST scans and 4 blocks of anatomical scans including a 3D T1 weighted scan, a 2D T1 mapping scan, a 2D T2 mapping scan, and a 2D B₁ mapping scan. **b** Comparison for liver Z-spectra and MTRasym spectra between Baseline and 51-min post-OGT, indicating the peak at 2.1ppm. **c** Representative dynamic glucose-weighted CEST images and the corresponding time course, for glucose CEST changes (MTRasym 2.1 ppm, all time points subtracted the initial value before glucose takeup, error bar: value standard deviation among all motion-stabilized liver voxels). **d** Same layout as (**c**), but for the corresponding T2 maps of the slice shown in (**c**). The time-courses for other subjects are displayed in Supplementary Fig. 6.1 and all-time-point images of the same subject are displayed in Supplementary Fig. 6.2, depicting the location of pancreas.

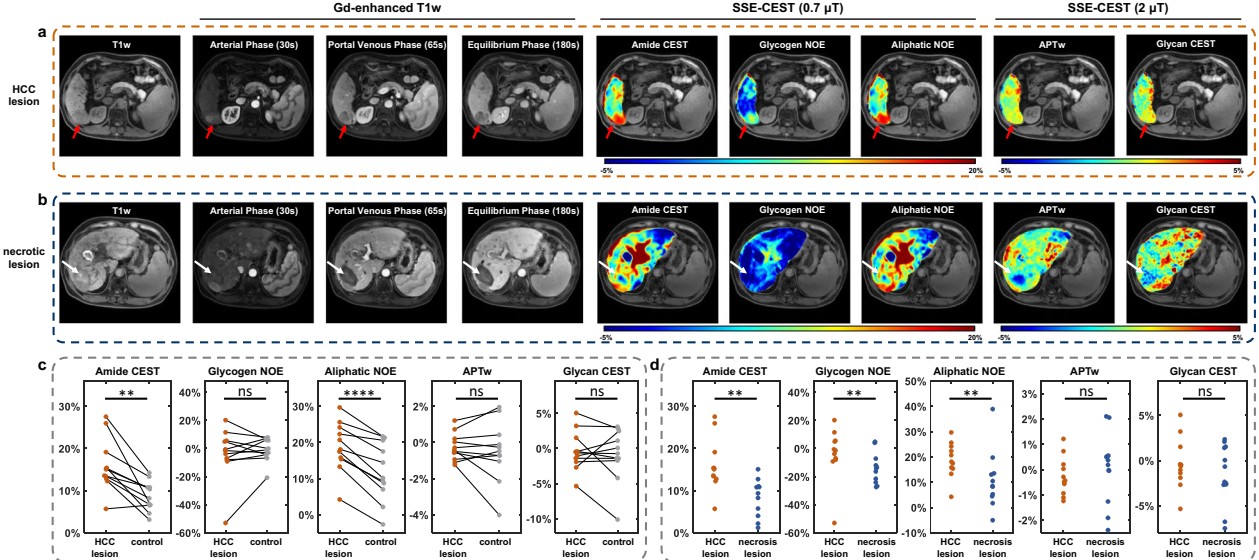

**Fig. 7 | 3D liver metabolic mapping in HCC lesions and necrotic lesions by SSE-CEST. a, b** Representative axial images of an active HCC lesion (**a**, red arrow) and a treated necrotic lesion (**b**, white arrow). For each case, the panel includes anatomical T1-weighted, contrast-enhanced (arterial, portal venous, and equilibrium phases), and SSE-CEST metabolic maps. CEST images comprise Lorentzian-difference (LD) maps derived from the 0.7 µT acquisition at +3.5 ppm (amide CEST), −3.5 ppm (aliphatic NOE), and −1.2 ppm (glycogen-related NOE), as well as MTR asymmetry maps from the 2 µT acquisition at 3.5 ppm (APTw) and 1.2 ppm (glycan CEST). Active HCC displays hyper-intensity on 0.7 µT LD maps, while the necrotic lesion is hypo-intense across all CEST contrasts. **c** Statistical comparison between active HCC lesions and paired control normal liver tissue ($n = 11$, two-sided paired t-test; Amide CEST $p = 0.0013$, Glycogen NOE $p = 0.5299$, Aliphatic NOE $p < 0.0001$, APTw $p = 0.5238$, Glycan CEST $p = 0.5646$). **d** Statistical comparison of CEST metrics between active HCC lesions ($n = 11$) and necrotic lesions ($n = 12$, two-sided unpaired t-test; Amide CEST $p = 0.0066$, Glycogen NOE $p = 0.0052$, Aliphatic NOE $p = 0.0052$, APTw $p = 0.5780$, Glycan CEST $p = 0.8751$). *, **, ***, **** denote $p < 0.05$, $p < 0.01$, $p < 0.001$, $p < 0.0001$, respectively.

acquired (Zpos-0.7 µT, Zneg-0.7 µT and Z-2 µT), each within a single breath-hold (Supplementary Table 1, Fig. 5.1). Visually, the active HCC lesion showed marked hyper-intensity on all three 0.7 µT LD maps (i.e., amide CEST, aliphatic NOE and glycogen-related NOE), indicating elevated levels of mobile proteins and altered macromolecular environment. The detailed spectra allowed further contrast interpretation and quality control, with the spectral bottom all centered at 0 ppm (Supplementary Fig. 7.4). In contrast, the necrotic lesion appeared hypo-intense across all CEST contrasts. All 41 slices of APTw and 0.7 µT ST(3.5 ppm) images demonstrated SSE-CEST's robust spatial reconstruction, revealing the complete lesion distribution. (Supplementary Figs. 7.2, 7.3). Interestingly, the APTw maps from 2 µT SSE-CEST did not show as obvious hyper-intensity contrast as those on 0.7 µT images.

Quantitative analysis corroborated these observations (Fig. 7). When comparing active HCC to necrotic tissue, the 0.7 µT LD signals for amide CEST, glycogen NOE, and aliphatic NOE were all significantly higher in HCC ($p = 0.0066$, $p = 0.0052$, and $p = 0.0052$, respectively). Similarly, compared to adjacent normal liver, active HCC exhibited significantly elevated LD signals for aliphatic NOE ($p = 0.000019$) and amide CEST ($p = 0.0013$). In contrast, none of the 2 µT MTR asymmetry metrics showed significant differences in these comparisons. This discrepancy may be attributed to the inherent property of asymmetry analysis: both the +3.5 ppm (amide) and −3.5 ppm (aliphatic NOE) pools can yield saturation effects. Their concurrent elevation in HCC might lead to substantial cancellation upon subtraction, thereby attenuating the sensitivity of the asymmetry metric (APTw) despite of genuine metabolic differences. These results demonstrate that the LD contrasts from the 0.7 µT acquisition offer superior specificity for differentiating active HCC from necrotic or normal tissue, underscoring the value of multi-B1, spectrally-resolved assessment with SSE-CEST. To clinically evaluate the detection of focal lesions using SSE-CEST, two radiologists independently contoured lesions on the ST maps in 21 patients while blinded to other imaging results. Inter-reader agreement was excellent (Cohen's kappa = 0.89). The lesion detection

rates were as follows: for active HCC lesions ($n = 11$), 100% and 90.9% for the two readers; for treated necrotic lesions ($n = 12$), 100% and 100% for the two readers. These detection rates align with the expected visibility of different tissue types on metabolic images and support the clinical reliability of SSE-CEST for lesion assessment. This analysis revealed excellent inter-reader agreement (Cohen's kappa = 0.89) and demonstrated clinically meaningful detection rates that logically correlated with expected metabolic activity: for active HCC lesions ($n = 11$), 100% and 90.9% for the two readers; for treated necrotic lesions ($n = 12$), 100% and 100% for the two readers; while benign lesions including cysts and hemangiomas ($n = 22$) exhibited lower rates of 45.5% and 50% for the two readers. These detection rates align with the expected visibility of different tissue types on metabolic images and support the clinical reliability of SSE-CEST for lesion assessment.

To validate metabolic specificity, we compared SSE-CEST with [18]F-FDG PET/CT in an HCC patient showing a PET hot spot 10 months post-treatment (Fig. 8). SSE-CEST contrast maps acquired 3-week after PET scan revealed matched hyper-intense regions (red arrows) on three representative axial slices. The SSE-CEST maps included the amide-specific LD contrast at 3.5 ppm acquired with a low saturation power (B1 = 0.7 µT), reflecting mobile protein/peptide content, and the glucose-weighted ST contrast at 2.1 ppm acquired with a higher power (B1 = 2 µT). Both contrasts also display adjacent necrotic regions (white arrows). This agreement with PET/CT and Gd-enhance MRI suggest the potential of SSE-CEST for 'label-free' non-radiative way in monitoring treatment response.

## Discussion

We have developed a generalized scheme for ultra-fast motion-stabilizing CEST-MRI, and demonstrated its clinical feasibility for differentiating benign liver necrosis from metabolic-active malignant ones. This spatial-spectral encoding (SSE) scheme presents four advantages. 1) achieved 3D abdominal CEST MRI within a single breath-hold, producing 'organ-frozen' images with minimal respiratory motion artifacts

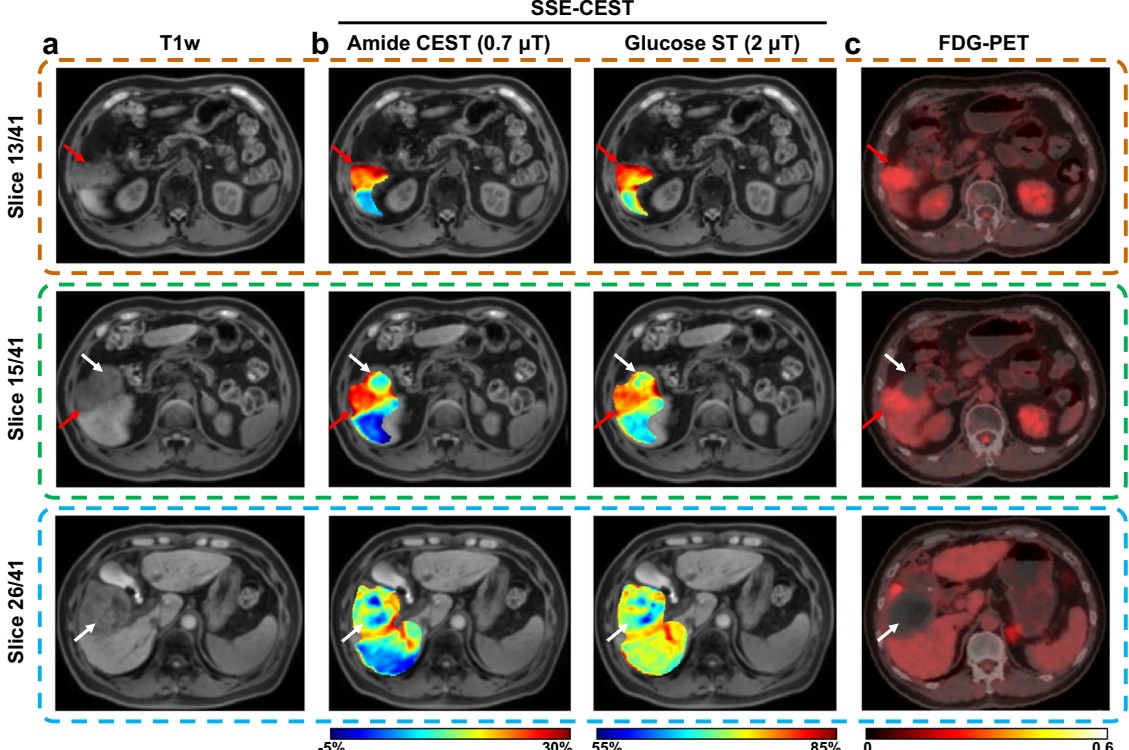

**Fig. 8 | Comparison of SSE-CEST metabolic maps and FDG-PET in a patient with hepatocellular carcinoma (HCC), scanned 10-month post treatment.** Three axial slices are shown, highlighting active HCC lesion (red arrows) and adjacent post-treatment necrotic regions (white arrows). **a** Anatomical T1 weighted image. **b** SSE-CEST metabolic maps: (left) amide CEST (LD (3.5 ppm)) using B1 of 0.7 μT, reflecting mobile protein/peptide content; (right) the ST image at 2.1 ppm (B1 of 2 μT), reflecting glucose-related metabolism. **c** Corresponding slices on the fused images of $^{18}$FDG-PET and CT. The hyper-intense regions on SSE-CEST images matched well with the 'hot-spot' regions on PET images, indicating the metabolic activeness.

and high repeatability. 2) enabled highly-efficient sampling of multiple saturation frequencies, allowing simultaneous detection for multiple kinds of metabolites (e.g., glycogen, glucose and mobile proteins), while improving correction of $B_0$ inhomogeneity and quantification accuracy. 3) Motion-free 3D images from a single breath-hold enabled precise spectral alignment and co-registration with high-resolution MRI datasets, facilitating CEST metabolic interpretation and multimodal diagnosis. 4) enabling capture of signal kinetics and biodistribution across multiple abdominal organs after exogenous contrast agent administration. SSE-CEST enables rapid, stable metabolic imaging on clinical 3.0 T MR scanners, providing non-invasive 3D metabolic mapping to transform diagnosis and prognosis of liver disease and other abdominal disorders.

SSE-CEST establishes a unified framework for concurrent spatial-spectral encoding and data-driven decoding exploiting spectral low-rankness, yielding flexible $z$–$\omega$ encoding patterns that integrate readily with acceleration strategies for spatial imaging. The basic unit of SSE-CEST acquisition is similar to previous ultra-fast Z-spectroscopy (UFZ), which turns on the gradient along slice direction (Gz) during the saturation pulse[25]. While a single UFZ sequence allows simultaneous labeling at multiple saturation $\omega$, previous works failed to fully resolve the 3D spatial distribution[25–27]. Our SSE framework considered the 4D CEST data as a $z$-$\omega$ plane plus a voxel dimension, and designed diagonal $z$-$\omega$ sampling pattern consisting of multiple UFZ scans with either varied Gs or $\omega_O$. SSE-CEST enable efficient sampling of the 4D space, substantially reducing the number of saturation-encoding steps required in conventional CEST (Fig. 1). SSE-CEST scheme could be easily combined with spatial acceleration strategies, including parallel imaging, compressed sensing, half scan, or keyhole methods. Here, by integrating parallel imaging and TFE-dixon readout with SSE-CEST, we achieved a total acceleration factor of 44×, enabling a full spectral scan

of whole-liver 3D images within a single breath-hold. (11 parallel $z$-$\omega$ lines for the 2 μT SSE-CEST protocol and 10 for the 0.7 μT ones). Another two types trajectories, diamonds and radial, were also demonstrated in ex vivo porcine, with the radial pattern only further cut 40% scan time (Fig. 2, Supplementary Fig. 2.1). For this initial technical validation in human abdominal imaging, we implemented parallel trajectory with constant $G_{sat}$ and uniform sampling interval of $\omega$ across slices, thereby avoiding artifacts and inconsistencies from gradient variations while ensuring uniform reconstruction and quantification. These findings demonstrate that SSE-CEST patterns can be customized and optimized for specific metabolites and anatomical regions, with potential extension to other MR spectroscopic techniques.

SSE-CEST achieved high-fidelity spatial-spectral reconstruction in glycogen phantoms and ex vivo porcine liver compared with conventional CEST acquisitions. Under multiple tested saturation $B_1$'s, SSE-CEST images exhibited well-preserved anatomical details (e.g., major vessels in porcine liver), while the spectra maintained excellent quantitative accuracy. Slice-by-slice comparison of phantoms between SSE-CEST and conventional CEST exhibited visually identical contrast, with strong linear correlations observed both between their signals and between SSE-CEST signals and glycogen concentrations. Noteworthy, different from conventional CEST, the saturation bandwidth of SSE-CEST is constrained by saturation gradient ($G_{sat}$) and slice thickness. For example, our human protocol obtained a saturation bandwidth of ~0.1 ppm, with $G_{sat}$ of 0.1 mT/m and 2.92 mm-thick slices. Although SSE-CEST grouped adjacent-slice voxels for accurately extracting spectral basis, it does not change the resolution on Z-dimension, either for the in-plane resolution, as evidenced both in ex vivo porcine liver (Supplementary Fig. 1.3) and in tumor patient (Supplementary Figs. 7.1, 7.5). SSE-CEST also displayed potential to

detect sub-centimeter lesions (<1 cm) using a higher-resolution scan protocol. (Supplementary Fig. 7.6).

SSE-CEST revealed its unique advantages in revealing endogenous metabolic changes from multiple abdominal organs, in a fasting experiment and in recognizing metabolic-active liver lesions in patients. Owing to the 3D imaging capability in single-breath hold time-scale, SSE-CEST enable 'motion-free' images with high repeatability. The well alignment among multi-offset images leads to reliable quantitation, as suggested by the five contrast maps in Fig. 5 and Fig. 7. Notably, the spectral-fitting analysis methods are now allowed (e.g., LD and R1ρ), improving the contrast interpretation and quality control. Fasting-state 0.7 μT SSE-CEST revealed glycogen NOE alterations and marked amide-CEST signal reductions. In cancer patients, full-spectrum SSE-CEST scans at 0.7 μT generated three LD contrast maps. Active HCC lesions exhibited significantly higher amide-CEST and aliphatic NOE signals compared to necrotic and normal-appearing regions within the same slice. However, glycogen NOE did not differ between active HCC and controls, suggesting altered glycogen metabolism may extend beyond active lesion regions. The 3D SSE-CEST images could easily fused and overlapped on a high-resolution T1w images, aiding for multi-contrast diagnosis (Figs. 4–8).

The application scenarios of SSE-CEST includes both the endogenous contrast and the kinetic imaging upon administration of exogeneous agents including natural glucose[11,30] and clinical approved CT agents for pH mapping[33]. In this first technical report, we employed the oral glucose take-up (OGT) in healthy volunteers for demonstration. The full-spectral 3D scanning capability of SSE-CEST allowed continuous monitoring of spectral changes over time, enhancing data interpretability and reliability of dynamic metabolic imaging. During OGTT, the most significant spectral changes in MTRasym and Z-spectra occurred at -2.1 ppm (Fig. 6b), which was subsequently employed for glucose metabolism imaging (Fig. 6 and Fig. 8). Beyond oral administration, SSE-CEST's ultrafast, motion-stabilized acquisition uniquely positions it to capture rapid kinetics of intravenously administered agents in oncology, potentially enabling real-time metabolic tumor profiling and treatment response monitoring.

Despite these advancements, several limitations must be addressed, especially for clinical applications. First, current protocol with the breath-hold duration of -18 s may be challenging for patients with respiratory difficulties. A possible solution is moving S0 scan to another breath-hold and carrying on registration in the post-processing stage. This will reduce the 0.7 μT scan to 18.7 seconds and the 2 μT protocol to 16 s. Besides, other trajectories like radial ones could be explored, and better readout acceleration strategies could be also applied. Second, the B1-inhomogeneity of a large field-of-view was not concerned in this study, which could affect the reliability of images especially for the edge regions and slices. In future B1-correction could be performed based on multi-B1 acquisition[34]. Third, the quality of extracted CEST maps may not as good as the raw ST images, due to the misalignment between images, inevitable motions from heart and other digestive organs, the in-perfect fat suppression and B0 correction. For example, the linear shift of ω in parallel trajectory result in a 'dead area' in edge slices, which may induce missing values in APTw images after B0-correction. In future, the alignments among encoding lines, and analysis for suppressing motion artifacts could be added. Finally, our single Lorentzian approximation for water saturation at 0.7 μT overlooks line broadening and magnetization transfer effects that become prevalent at higher B₁ amplitudes. Extension to Gaussian, Voigt, or multi-pool lineshapes is warranted to ensure fidelity across diverse saturation conditions and tissue types.

In summary, we have demonstrated the capabilities of a single-breath-hold CEST approach for metabolic measurement in human livers, observing 3D spatial distribution of multiple metabolites, including glucose, glycogen, proteins, and lipids. The full-spectral motion-free SSE-CEST technique may push a step forward for future metabolic characterization of liver diseases and other abdominal disorders.

## Methods

### Acquisition and reconstruction

**Single readout of 3D images with saturation-gradient.** Similar to the UFZ sequence, we employ a saturation labeling strategy by simultaneously applying a $z$-direction gradient with frequency-selective saturation RF pulse centered at $\Delta\omega$. This allows simultaneous acquisition of spatial and spectral information in a single 3D volume. The relationship between the spatial location and the frequency offset is defined in Eq.(1). To achieve faster readouts, a 3D single-shot Turbo-field-echo (TFE) readout was employed, with k-space ordering processing from lower to higher frequencies. The proposed framework incorporates 3-echo Dixon readout to separate water and fat signal and obtain a B0 map. A $z$-$\omega$ encoding block and a readout block constitute a single imaging unit.

**Multiple readouts with designated $z$-$\omega$ trajectory.** To recovery the 4D tensor, the imaging unit is repeated multiple times. The 3D data from a single imaging unit constitutes a 3D plane in $x$-$y$-$z$-$\omega$ space, of which the location in 4D space was determined by $G_{sat}$ and $\omega_0$. The repeated acquisition with varied $G_{sat}$ and $\Delta\omega$ forms a specific $z$-$\omega$ trajectory. Especially, if $G_{sat}$ is fixed to 0, this imaging protocol are identical to the conventional CEST sequence. Notice, the multiple readouts were completed in a single breath-hold in our implementation.

**4D CEST-spectra reconstruction with $z$-$\omega$ decoding.** Reconstruct of the 4D CEST-spectra is achieved using a $z$-$\omega$ decoding strategy that exploits the anatomical and spectral similarity within a single slice or adjacent slices ($N_{adj} = 0, 1, 2, \ldots$). B0-correction was performed immediately after the interpolation step, using the Dixon-derived B0 map. Consider the $z^{th}$ slice with $m$ voxels and $n$ acquired frequency offsets:

$$\Omega^z = \left[ \omega_1^{(z)}, \omega_2^{(z)}, \cdots, \omega_n^{(z)} \right]^{T} \tag{2}$$

Due to the existence of $z$-$\omega$ encoding gradient, the two adjacent slices sample distinct frequency offsets:

$$\Omega^{(z-1)} = \left[ \omega_1^{(z-1)}, \omega_2^{(z-1)}, \cdots, \omega_n^{(z-1)} \right]^{T}, \Omega^{(z+1)} = \left[ \omega_1^{(z+1)}, \omega_2^{(z+1)}, \cdots, \omega_n^{(z+1)} \right]^{T}$$
$$\text{s.t.} \left| \omega_k^{(z-1)} - \omega_k^{(z)} \right| = \left| \omega_k^{(z+1)} - \omega_k^{(z)} \right| = \gamma \left| G_{sat\,k} \right| \Delta z, k = 1, 2, \cdots, n \tag{3}$$

For $z^{th}$ slice, the unresolved raw signal could be expanded into a matrix of $n \times m$, $\mathbf{S}^z$, where $n$ is the number of $z$-$\omega$ encodings and $m$ is the voxel numbers. To combine slices with distinct sampled $\Omega$, we initialized a unified zero matrix of size $m \times N\omega_{full}$ (the number of fully-sampled $\omega$ with 0.1 ppm interval), and filled the acquired values in corresponding $\omega$ position. Taking $N_{adj} = 3$ (using one adjacent slice on each side), we pack signals from three consecutive slices as $\mathbf{S_{raw}} = [\mathbf{S}^{z-1} \mathbf{S}^z \mathbf{S}^{z-1}]$ to extract the spectral basis, with a matrix format in below:

$$\mathbf{S_{raw}} = \left[ \mathbf{S}^{z-1} \mathbf{S}^z \mathbf{S}^{z-1} \right] = \begin{bmatrix} s_{\omega_1^{(z-1)},1} & \cdots & s_{\omega_1^{(z-1)},m} & 0 & \cdots & 0 & 0 & \cdots & 0 \\ 0 & \cdots & 0 & s_{\omega_1^{(z)},1} & \cdots & s_{\omega_1^{(z)},m} & 0 & \cdots & 0 \\ 0 & \cdots & 0 & 0 & \cdots & 0 & s_{\omega_1^{(z+1)},1} & \cdots & s_{\omega_1^{(z+1)},m} \\ \vdots & \vdots & \vdots & \vdots & \ddots & \vdots & \vdots & \vdots & \vdots \\ s_{\omega_n^{(z-1)},1} & \cdots & s_{\omega_n^{(z-1)},m} & 0 & \cdots & 0 & 0 & \cdots & 0 \\ 0 & \cdots & 0 & s_{\omega_n^{(z)},1} & \cdots & s_{\omega_n^{(z)},m} & 0 & \cdots & 0 \\ 0 & \cdots & 0 & 0 & \cdots & 0 & s_{\omega_n^{(z+1)},1} & \cdots & s_{\omega_n^{(z+1)},m} \end{bmatrix} \tag{4}$$

Performing SVD to $\mathbf{S_{raw}}$, resulting in $\mathbf{S_{raw}} = \mathbf{U} \cdot \boldsymbol{\Sigma} \cdot \mathbf{V}'$, where $\mathbf{U}$ is the spatial coefficient matrix, $\boldsymbol{\Sigma}$ is the diagonal matrix of singular values in descending order, $\mathbf{V}$ is the right singular matrix, with the matrix form of

$$\mathbf{V} = \begin{bmatrix} \mathbf{V}_{\omega_1^{(z-1)},1} & 0 & 0 & \cdots & \mathbf{V}_{\omega_1^{(z-1)},n} & 0 & 0 \\ 0 & \mathbf{V}_{\omega_1^{(z)},1} & 0 & \cdots & 0 & \mathbf{V}_{\omega_1^{(z)},n} & 0 \\ 0 & 0 & \mathbf{V}_{\omega_1^{(z+1)},1} & \cdots & 0 & 0 & \mathbf{V}_{\omega_1^{(z+1)},n} \\ \vdots & \vdots & \vdots & \ddots & \vdots & \vdots & \vdots \\ \mathbf{V}_{\omega_n^{(z-1)},1} & 0 & 0 & \cdots & \mathbf{V}_{\omega_n^{(z-1)},n} & 0 & 0 \\ 0 & \mathbf{V}_{\omega_n^{(z)},1} & 0 & \cdots & 0 & \mathbf{V}_{\omega_n^{(z)},n} & 0 \\ 0 & 0 & \mathbf{V}_{\omega_n^{(z+1)},1} & \cdots & 0 & 0 & \mathbf{V}_{\omega_n^{(z+1)},n} \end{bmatrix} \quad (5)$$

Note that the submatrix of $\mathbf{V}$, $\mathbf{v}^{(z)} = \left[ v_{\Omega^{(z)},1}, v_{\Omega^{(z)},2}, \cdots, v_{\Omega^{(z)},n} \right]$, is the spectral basis matrix of the $z^{th}$ slice. Due to the anatomical and similarity of the adjacent slices, we assume that $\mathbf{v}^{(z-1)}, \mathbf{v}^{(z)}, \mathbf{v}^{(z+1)}$ are three submatrices of $\mathbf{v}_{full}^{(z)}$, that derived from the fully-sampled spectra of 0.1ppm. Thus, the denser spectral bases of the $z^{th}$ slice can be approximated by combining $v^{(z-1)}$, $v^{(z)}$, and $v^{(z+1)}$:

$$\mathbf{V_C} = \begin{bmatrix} \mathbf{V}_{\omega_1^{(z-1)},1} & \mathbf{V}_{\omega_1^{(z-1)},2} & \cdots & \mathbf{V}_{\omega_1^{(z-1)},n} \\ \mathbf{V}_{\omega_1^{(z)},1} & \mathbf{V}_{\omega_1^{(z)},2} & \cdots & \mathbf{V}_{\omega_1^{(z)},n} \\ \mathbf{V}_{\omega_1^{(z+1)},1} & \mathbf{V}_{\omega_1^{(z+1)},2} & \cdots & \mathbf{V}_{\omega_1^{(z+1)},n} \\ \vdots & \vdots & \ddots & \vdots \\ \mathbf{V}_{\omega_n^{(z-1)},1} & \mathbf{V}_{\omega_n^{(z-1)},2} & \cdots & \mathbf{V}_{\omega_n^{(z-1)},n} \\ \mathbf{V}_{\omega_n^{(z)},1} & \mathbf{V}_{\omega_n^{(z)},2} & \cdots & \mathbf{V}_{\omega_n^{(z)},n} \\ \mathbf{V}_{\omega_n^{(z+1)},1} & \mathbf{V}_{\omega_n^{(z+1)},2} & \cdots & \mathbf{V}_{\omega_n^{(z+1)},n} \end{bmatrix} \quad (6)$$

As seen, grouping $N_{adj}$ adjacent slices could resolve a combined spectral bases matrix, $\mathbf{V_C}$, enabling $N_{adj}$-fold spectral sampling rate.

$$\min_{\mathbf{U}} \left\| \mathbf{M} \cdot \left( \mathbf{U} \mathbf{V_C}^{\mathbf{T}} - \mathbf{S_{corr}} \right) \right\|_2^2 \quad (7)$$

where $\mathbf{U}$ is the spatial coefficient matrix, $\mathbf{S_{corr}}$ is the B0-corrected signal of the $z^{th}$ slice, with each voxel shifted by its B0-drift value along $\omega$-axis, and $\mathbf{M}$ is a binary matrix that describes the acquired offsets after B0 correction. After minimizing Eq. (7) using gradient decent algorithm, the reconstructed SSE value can be calculated by:

$$\mathbf{S_{recon}} = \mathbf{U} \mathbf{V_C}^{\mathbf{T}} (8) \quad (8)$$

where $\mathbf{S_{recon}}$ is the reconstructed $S$-value matrix of the $z^{th}$ slice. For slices located on the edge of FOV, adjacent slices were selected only on the single side within the FOV. For these slices, sampling rate of spectral bases decreased due to less adjacent slices.

## Ex vivo validation of SSE-CEST

**Porcine liver experiments.** A whole, fresh porcine liver (never frozen) was procured from a commercial slaughterhouse and underwent MR scans within four hours post-mortem. The liver was placed in a glass box during MR scanning. MR protocol for porcine liver consists of sequences of four $z$-$\omega$ trajectories: (1) 'parallel' pattern (30 encodings). $\Delta\omega(ppm)|G_{sat}(mT/m)$ was: $-1560|0$, $-10|-0.1$, $-8|-0.1$, $-7|-0.1$, $-6|-0.1$, $-5|-0.1$, $-4.5|-0.1$, $-4|-0.1$, $-3.5|-0.1$, $-3|-0.1$, $-2.5|-0.1$, $-2|-0.1$, $-1.5|-0.1$, $-1|-0.1$, $-0.5|-0.1$, $0|-0.1$, $0.5|-0.1$, $1|-0.1$, $1.5|-0.1$, $2|-0.1$, $2.5|-0.1$, $3|-0.1$, $3.5|-0.1$, $4|-0.1$, $4.5|-0.1$, $5|-0.1$, $6|-0.1$, $7|-0.1$, $8|-0.1$, $10|-0.1$; (2) 'diamond' pattern (30 encodings). $\Delta\omega(ppm)|G_{sat}(mT/m)$ was: $-1560|0$, $-10|-0.1$, $-10|0.1$, $-7|-0.1$, $-7|0.1$, $-5|-0.1$, $-5|0.1$, $-4|-0.1$, $-4|0.1$, $-3|-0.1$, $-3|0.1$, $-2|-0.1$, $-2|0.1$, $-1|-0.1$, $-1|0.1$, $0|-0.1$, $0|0.1$, $1|-0.1$, $1|0.1$, $2|-0.1$, $2|0.1$, $3|-0.1$, $3|0.1$, $4|-0.1$, $4|0.1$, $5|-0.1$, $5|0.1$, $7|-0.1$, $7|0.1$, $10|-0.1$,

$10|0.1$; (3) 'radial' pattern (18 encodings). $\Delta\omega(ppm)|G_{sat}(mT/m)$ was: $-1560|0$, $-6|-0.6$, $-6|0.6$, $-4|-0.4$, $-4|0.4$, $-2|-0.2$, $-2|0.2$, $-1|-0.1$, $-1|0.1$, $0|0$, $1|-0.1$, $1|0.1$, $2|-0.2$, $2|0.2$, $4|-0.4$, $4|0.4$, $6|-0.6$, $6|0.6$; and (4) conventional CEST pattern (88 frequency offsets), in which saturation encoding gradient was turned off, but more frequency offsets: $-1560$, $-10:0.5:-5.5$, $-4.95:0.15:4.95$, and $5.5:0.5:10$ ppm. The MR protocol was repeated at three saturation power: 0.7 μT (3.5 s), 1 μT (3.5 s) and 2 μT (1.75 s). The 3D single-shot turbo-field-echo (TFE) readout parameters were: shot interval of 8000 ms, TE of 1.5/3.0/4.5 ms, flip angle of 15°, TR of 10 ms, SENSE of $2 \times 1$, resolution of $2 \times 2 \times 5.84$ mm$^3$, FOV of $230 \times 115 \times 61$ mm$^3$, echo train length of 302. The scan time for the conventional CEST protocol (scheme 4) was 11 min and 58 s, while the scan times of three SSE-CEST patterns are: parallel (4 min 14 s, 30 $z$-$\omega$ lines); diamond (4 min 22 s, 31 $z$-$\omega$ lines); radial (2 min 38 s, 18 $z$-$\omega$ lines). B0 maps were calculated from three-point Dixon readouts. Data from schemes 1-3 were compared with the conventional CEST data (scheme 4) using Bland-Altman analysis, to evaluate the consistency of the SSE-CEST with the gold-standard method.

**Glycogen phantom experiments.** The phantom comprises six 50 mL tubes, each containing oyster glycogen (Sigma-Aldrich, G8751) dissolved in PBS (pH 7.4) at concentrations of 0 mM, 6.25 mM, 12.5 mM, 25 mM, 50 mM, and 100 mM. For SSE-CEST protocol, $G_{sat}$ was fixed to $-0.1$ mT/m, and frequency offsets (a total of 30) were: $-1560$, $-10$, $-8$, $-7$, $-6$, $-5$, $-4.5$, $-4$, $-3.5$, $-3$, $-2.5$, $-2$, $-1.5$, $-1$, $-0.5$, $0$, $0.5$, $1$, $1.5$, $2$, $2.5$, $3$, $3.5$, $4$, $4.5$, $5$, $6$, $7$, $8$, and $10$ ppm. For conventional CEST protocol, frequency offsets (a total of 88) were: $-1560$, $-10:0.5:-5.5$, $-4.95:0.15:4.95$, and $5.5:0.5:10$ ppm. The 3D TFE readout parameters were: shot interval of 10000 ms, TE of 1.03 ms, flip angle of 10°, TR of 5 ms, SENSE of $2 \times 1$, resolution of $2 \times 2 \times 3$ mm$^3$, FOV of $157 \times 157 \times 61$ mm$^3$, echo train length of 402.

**Data analysis.** For porcine liver, the reconstructed images were additional corrected for B0-inhomogeneity using the lowest point as water frequency (Self B0-correction), and ST images were obtained by 1-Z along the entire frequency dimension ($-10$ to $10$ ppm, resolution 0.1 ppm). LD images were calculated for amide (3.5 ppm), glycans ($-1.2$ ppm), and aliphatic ($-3.5$ ppm). For a clarified visualization, an $8 \times 8$ window was used to down-sample the ST images from matrix $64 \times 128$ to $8 \times 16$. A non-zero array was generated for each contrast with each acquisition trajectory after averaging the values in each voxel and removing zeros from the voxels. Afterwards, Bland-Altman analysis was used to test the consistency of spatial performance between the $z$-$\omega$ acquisition trajectories and the conventional one for all the contrasts. The data was evaluated using bias (mean ± std) and 95% Limits of Agreement. To test statistical significance, normal distribution of all arrays was tested first. Then Mann Whitney tests were performed between the $z$-$\omega$ acquisition trajectories and the conventional one for all the contrasts. For spectral evaluation, a representative ROI was selected from the down-sampled T1-weighted image in the same slice. Averaged CEST-spectra were calculated within this ROI for all the trajectories. Root mean square error (RMSE) along the whole CEST-spectra between three $z$-$\omega$ acquisition trajectories and the conventional one was calculated to verify the spectral accuracy.

For glycogen phantom, LD-0.7μT images of glycogen phantoms 1.2 ppm and -1.2 ppm were generated. For comprehensive statistics and better SNR, values within each tube in each slice were averaged, generating $6 \times 11$ matrixes for LD 1.2 ppm and ST -1.2 ppm, respectively. Linear regression associated with Pearson correlation was used to evaluate the relationship between LD values and glycogen concentration. Each data presented as mean ± SD represented data of a tube from all slices. To verify the consistency between our method and the gold standard, linear regression associated with Pearson correlation was performed on all the 66 points.

## In vivo validation of SSE-CEST

**Human participants.** In vivo MRI experiments on healthy volunteers were performed at the Center for Biomedical Imaging Research, Tsinghua University, approved by the Institutional Review Board of Tsinghua University. Patient scans were performed at Beijing Tsinghua Changgung Hospital, approved by the Institutional Review Board of Beijing Tsinghua Changgung Hospital. All participants provided written informed consent before MRI scans.

**MRI protocols.** All MR experiments were conducted on 3-Tesla Philips Ingenia CX scanners (Philips Healthcare, Best, The Netherlands), using a 16-channel body coil and a 16-channel posterior coil as receivers. For all subjects, two parallel-type CEST labeling schemes were implemented: (1) B1 of 2 μT, saturation time of 0.4 s, 12 saturation encodings with frequency offset of -1560, −6, −4.8, 4.8, −3.6, 3.6, −2.4, 2.4, −1.2, 1.2, 0 ppm. Shot interval was 1472 ms. $G_{sat}$ was fixed to −0.1 mT/m. (2) B1 of 0.7 μT, saturation time of 0.8 s, 11 saturation encodings, with frequency offset of −1560, −9, −6, −5, −4, −3, −2, −1, 0, 1.5, 3 ppm for the NOE contrast acquisition (Zneg), and with frequency offset of −1560, 9, 6, 5, 4, 3, 2, 1, 0, −1.5, −3 ppm for the CEST contrast detection (Zpos). Shot interval was 1871 ms. $G_{sat}$ was fixed to −0.1 mT/m. The Dixon-type 3D turbo-field-echo (TFE) readouts were employed (Supplementary Fig. 4.1): voxel size=4.5 × 4.5 × 2.9 mm$^3$, field of view (FOV) = 320 × 288 × 150 mm$^3$, flip angle (FA) = 8°, repetition time (TR) = 2.9 ms, echo time (TE) = 0.77/1.27/1.77 ms, echo train length=350. Each sequence was performed in single breath-hold.

**Repeatability test.** Ten healthy volunteers were recruited (eight males and two females; age of 24.8 ± 3.3 years), and scanned twice using MRI protocols above. Repeatability of SSE-CEST was evaluated using Pearson's correlation coefficients, and Bland-Altman analysis.

**Fasting experiments.** Data were acquired in five subjects (four males and one female; age of 24.2 ± 1.5 years). For each subject, two scans were performed: (1) a post-meal scan on the evening (20:00), which was performed 2 hours after a full dinner, and (2) a post-fasting scan in the following morning (08:00), which was performed after overnight fasting (~12 h after the post-meal scan). Participants have been instructed to have nothing to eat or drink other than water in the intervening period. Data from the two scans were reconstructed and post-processed independently. Differences between the post-meal scan and the post-fasting scan were measured using two-sided Student's t-tests.

**Oral glucose test.** Five healthy volunteers (four males and one female; age of 24.8 ± 1.92 years) were recruited for oral glucose test experiments. Each subject underwent an MRI scan that begins at 7:30 a.m. after overnight fasting. The MRI scan consists of four stages: (1) Before OGT: a pre-scan and an SSE-CEST. (2) OGT: the subject drank 300 ml of sugar water, in which 75 g of anhydrous glucose was dissolved. (3) After OGT (0–21 min): a pre-scan and six repeated SSE-CEST, with time interval of five minutes. (4) After OGT (26–51 min): a pre-scan and six repeated SSE-CEST, with time interval of five minutes. A total of 13 SSE-CEST data was acquired and reconstructed. ST signal at 2 ppm, MTR asymmetry signal at 2 ppm, and line-width of CEST-spectrum were plotted over time.

**Liver patients.** Twenty-one patients with focal liver lesions were prospectively enrolled, comprising 8 benign lesions and 12 hepatocellular carcinomas (HCCs). Among the HCC cohort, 11 were post-treatment. One HCC patient was excluded from statistical analysis due to protocol deviation for technical validation.

Two radiologists (Y.Y. and B.Z.) independently contoured lesions on CEST images; perilesional normal parenchyma served as control regions on the same slice. Lorentzian difference (LD) and magnetization transfer ratio (MTR) asymmetry values were quantified by averaging voxels within each ROI.

## Data Analysis

Data reconstruction and CEST quantification was performed in MATLAB R2021b on a workstation. The reconstruction time was ~260 seconds. The CEST images and spectra were quantified by calculating MTR asymmetry (MTR$_{asym}$), Saturation Transfer values (ST), and Lorentzian Difference (LD), as defined by the following equations:

$$\text{MTR}_{asym} = \frac{S(-\omega_{CEST}) - S(\omega_{CEST})}{S_0} \tag{9}$$

$$\text{ST} = 1 - \frac{S(\omega_{CEST})}{S_0} \tag{10}$$

$$\text{LD} = Z_{CEST}^{LF} - Z_{CEST}^{exp} \tag{11}$$

where $\omega_{CEST}$ is the frequency offset, $S(\omega_{CEST})$ is the saturation image at $\omega_{CEST}$, $S_0$ is the reference image at -1560 ppm, $Z_{CEST}^{LF}$ is the fitted Lorentzian line-shape, and $Z_{CEST}^{exp}$ is the acquired CEST-spectrum.

## Reporting summary

Further information on research design is available in the Nature Portfolio Reporting Summary linked to this article.

## Data availability

The main data supporting the results in this study are available within the paper and its Supplementary Information. The raw data and quantification results from a representative participant in the fasting experiments are available at https://doi.org/10.5281/zenodo.18795840. Source data are provided with this paper.

## Code availability

The SSE-CEST reconstruction code is available at https://doi.org/10.5281/zenodo.18795840, the analysis code is available at http://github.com/easycest/SSE-CEST.

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

## Acknowledgements

This work was supported by National Key R&D Program of China (2022YFC3602500, X.S.), National Natural Science Foundation of China (12271434, X.H.), the Natural Science Basic Research Plan in Shaanxi Province of China (2023-JC-JQ-57, X.H.), stipend to C.L. from Jingjinji National Center of Technology Innovation, and Tsinghua University Initiative Research Program to X.S.; We thank Kaixiang Li for assistance with figure preparation and data organization.

## Author contributions

C.L., N.G., and X.S. conceived the project, designed the experiments and wrote the manuscript; X.S. supervised the study. C.L. developed the acquisition and reconstruction technique. C.L. and N.G. performed all the MRI experiments. H.R., H.L., J.H., Z.L., and X.H. assisted with data analysis. B.Z., Y.Y., and Z.Z. recruited the patients and performed clinical diagnoses. All authors discussed the results and approved the final manuscript.

## Competing interests

The X.S., C.L., and N.G. have filed a patent application on the SSE-CEST method (PCT/CN2023/125056), The other authors declare no competing financial interests.
