## [Transparent Peer Review file · Nature Communications]

Single-breath-hold 3D abdominal metabolic MRI enables label-free diagnosis of liver cancer

Corresponding Author: Professor Xiaolei Song

Version 0:

Reviewer comments:

Reviewer #1

(Remarks to the Author)

This work describes a new MRI acquisition technique for chemical exchange saturation transfer (CEST). CEST is a type of MRI contrast that is more specific for detecting molecular signals compared to other more conventional MRI scans such as T1 and T2 weighted imaging, and magnetization transfer contrast (MTC). CEST MRI has the potential to improve clinical care by elucidating the underlying mechanisms of several diseases, including cancer. The work presented here shows substantial improvement in the acquisition by acquiring a whole Z-spectrum within a single-breathhold which is important for imaging organs in the abdomen. Still, it is hard to judge the quality of the data using the new method via the analysis methods used here. As mentioned, CEST aims to specifically detect molecules. Acquiring a full Z-spectrum in this sense is important because it allows for the separation of different signal contributions using multiple-pool modeling approaches. The analysis methods used in this manuscript often report the signal change of water (ST) which contains much larger contributions from water relaxation and immobile macromolecules, thus confounding the specific signals from solute molecules. If the ST signal is sufficient, other approaches are already available for reporting this in a single breath hold. I believe the work is of interest to the readers of this journal and I would recommend publication after major revisions to the data analysis methods used to quantify reproducibility and CEST signal. Specific comments are listed below.

1. Since the focus of this paper is on CEST, which aims to extract Z-spectral peaks from background signals (e.g., MTC and direct water saturation), I believe the ST metric shouldn't be used. In Figure 2, the regional contrast from amides and glycogen is very similar which indicates larger contributions from water properties and MTC. Can the authors quantify the separate signals using multiple-pool fitting or background curve fitting via R1rho theory? Additionally, for validation, can the authors compare their maps with other non MR techniques?

2. In the difference spectra shown in Figure 2c, f, i, it is clear that the z-omega pattern does affect Z-spectral features. The Bland-Altman plots shown in Figure S7 are comparing whole Z-spectra which are primarily determined by water properties and MTC. CEST signals tend to be much smaller (approx 1-3%). It would be more important to compare extracted CEST signals with the different sampling schemes.

3. In Figure 2, it was determined that a parallel trajectory was the most reproducible. Can the authors explain why this was? Is it possible that it is due to only the frequency offset being changed with no changes in the gradient?

4. In Figure 3, and similar to before, I think it is incorrect to use the ST metric to characterize the glycan and glycogen signals. Background fitting or multiple-line fitting should be used.

5. What is causing the variation in contrast across the slices and within the tubes in Figure 3? I am assuming the mixture is homogeneous.

6. The methods section states that human experiments were performed using either 2 uT or 0.7 uT. The 2uT results shown in Figures 4 and 5 show that almost 90% of the signal was saturated at 1.2/-1.2 ppm. What did the 0.7 uT results look like? Did they have higher SNR? Also, it is unclear what is being reported on at these frequencies.

7. It is sometimes difficult to follow the signals reported and what they are meant to represent. For instance, for healthy volunteers, amides are at 3.5 ppm, amines from 2-3 ppm, and hydroxyls from 0.6 to 2 ppm. In the subsequent results

involving fasting, signals are reported at 3.5, 2, 1.2, -1.2, and -3.5 ppm. Is 2 ppm supposed to be amines or hydroxyls? In the glucose ingestion experiments, MTRAsym is reported in the main document and also linewidth in the supplementary. The parameters keep on changing throughout the paper and are hard to follow. I would like to see a consistent parameter used and one that is as specific to the molecular origin of the contrast.

8. In Figure 5, there is a statistically significant signal change for ST(3.5ppm) but not APTw. Does this indicate the changes are mostly due to changing water parameters?

9. Comparing the results in Figure 6, and Part 7 of the supplementary, it is not clear if the time course in Figure 6b is reproducible across the five subjects. I suspect the SNR is insufficient to measure changes due to glucose ingestion. In addition, given the relatively slow time course of glucose signal changes, is this rapid imaging technique suitable for this application?

10. Similar to Figure 5, there is a significant difference in the ST(3.5ppm) results but no significance in the APTw. Are the results due to changes in water parameters or MT instead of a CEST effect?

11. In some figures, it is not clear how many adjacent slices/points were used in the SVD. Using adjacent slices in SVD requires anatomical similarity. For heterogeneous regions (e.g., liver lesions) how would this affect the results?

Other comments:

12. In the supplementary, can you please describe how the resolution is calculated in z-omega space?

13. There is inconsistent use of delta omega (e.g., Figure 1d and Eq. 1)

14. I assume the total scan time is calculated by the number of saturation encodings multiplied by the shot interval. Based on the methods, this will mean a scan time of approximately 20 seconds. Can the authors clarify this?

Reviewer #2

(Remarks to the Author)

This is very interesting research, combining UFZ and low rank tensor-based reconstruction to achieve super-fast abdominal CEST imaging within a single breath-hold. The study was well-planned, including optimization of different z- ω encoding patterns, validation in glycogen phantoms, and in vivo studies in healthy subjects, fasting, glucose tolerance tests, and hepatic carcinoma patients (both HCC and necrotic lesions). Imaging framework is well presented, manuscript is very well prepared, and results are convincing. This technique has great potential of making CEST imaging feasible in the abdomen. I have a few questions:

- The major technical innovation is z- ω encoding, allowing x11 acceleration. It relies on interpolation of the spectral basis from adjacent slices, assuming the CEST signals are similar. It could be ok for normal subjects or diffuse disease but may have problems when there are metabolite concentration variations or focal disease. Zcorr was solved directly from the interpolated spectral basis without further refinement. The reliability of this approach needs to be rigorously tested.
- Are saturation times of 0.4 and 0.8s sufficient for the CEST signal to reach steady state? Usually 2-4s are used for continuous saturation.
- There was discrepancy between phantom and in vivo studies. In human studies, B1 = 2 uT with 0.4 s saturation and B1 = 0.7 uT with 0.8 s saturation, with TFE shot intervals of 1472/1871 ms. By contrast, in phantom studies, 0.7 uT (3.5 s), 1 uT (3.5 s), and 2 uT (1.75 s) were used with a TFE shot interval of 8 s. It would have been helpful if they had included phantom experiments using the same parameters as in vivo, and compared them with a conventional CEST spectrum.

Version 1:

Reviewer comments:

Reviewer #1

(Remarks to the Author)

The authors have done a commendable job addressing my previous comments. I feel that the changes make the research much stronger and it is suitable for publication a minor edit.

1. The broad direct water saturation line is not Lorentzian in nature and thus is probably the major reason for the poor fits in Supplementary Figures 5.1-5.3. Perhaps the authors can briefly mention in the Discussion or Section 5 of the supplementary that Gaussian lineshapes or multiple lineshape fitting may improve the specificity of the signal extracted.

Reviewer #2

(Remarks to the Author)

All my comments have been satisfactorily addressed. Congratulations on this important work!

Single-breath-hold 3D abdominal metabolic MRI: label-free imaging for enhanced liver disease diagnosis

(Manuscript ID: NCOMMS-25-48838-T)

We would like to thank the reviewers and editors for their time and insightful comments. **We're very appreciated that the reviewers gave us very positive feedbacks**, considering our work “show substantial improvement in the acquisition ... which is important for imaging organs in the abdomen”, “very interesting research”, “... results are convincing”, “has great potential of making CEST imaging feasible in the abdomen.” **Building on the reviewers' constructive guidance, we substantially refined our data analysis and reliability validations.** Three key improvements include:

1. **Quantitative analysis confirms the full-spectral, motion-stabilized benefits of SSE-CEST:** Replaced previous less-specific ST images in the main text with the extracted CEST imaging (except Fig. 4, used for repeatability), with five contrast maps defined consistently throughout; Statistical plots have been updated to match (**revised Figs. 2, 3, 5, 6, 7, Supplementary Table 1, Supplementary Methods 2, 3**).

2. **Reliability validated in phantoms and in ex vivo porcine liver:** Since in vivo scans lack motion-free gold standard, we have comprehensively compared SSE-CEST with the conventional CEST, in ex vivo porcine liver and in phantoms (re-prepared phantoms by dissolving glycogen in agar to mimic human tissue). All-slice images and reconstruction errors for 1-21 slice groups are now supplied (**Supplementary Figs. 2.1-2.3, 3.1-3.5**).

3. **Spatial-spectral performance in volunteers and in patients:** we now supply detailed spectra, whole-volume images and grouped-slice comparisons (**revised Figs. 5, Supplementary Figs. 5.1-5.7, 7.1-7.5**). Retrospective data enable direct matching of SSE-CEST images to ^{18}F -FDG-PET hotspots (**Fig. 8**), and prospective high-resolution acquisition in a further patient confirms the ability for focal-lesion detection (**Supplementary Fig. 7.6**).

We have also sharpened the discussion and reordered the Supplementary Information to mirror the main-text experiments, reinforcing reliability and quantification. We believe the ultra-fast motion-stabilizing SSE-CEST will significantly advance the diagnosis for liver cancer and other abdominal diseases, offering a label-free, clinical-accessible metabolic MRI solution.

The detailed corrections and responses to the reviewers' comments are listed below point-by-point.

Point-to-Point Response to Comments

Reviewer 1

Overall Comments: This work describes a new MRI acquisition technique for chemical exchange saturation transfer (CEST). CEST is a type of MRI contrast that is more specific for detecting molecular signals compared to other more conventional MRI scans such as T1 and T2 weighted imaging, and magnetization transfer contrast (MTC). CEST MRI has the potential to improve clinical care by elucidating the underlying mechanisms of several diseases, including cancer. **The work presented here shows substantial improvement in the acquisition by acquiring a whole Z-spectrum within a single-breathhold** which is important for imaging organs in the abdomen. Still, it is hard to judge the quality of the data using the new method via the analysis methods used here. As mentioned, CEST aims to specifically detect molecules. Acquiring a full Z-spectrum in this sense is important because it allows for the separation of different signal contributions using multiple-pool modeling approaches. The analysis methods used in this manuscript often report the signal change of water (ST) which contains much larger contributions from water relaxation and immobile macromolecules, thus confounding the specific signals from solute molecules. If the ST signal is sufficient, other approaches are already available for reporting this in a single breath hold. **I believe the work is of interest to the readers of this journal and I would recommend publication after major revisions to the data analysis methods used to quantify reproducibility and CEST signal.** Specific comments are listed.

[Comment 1.1]

Since the focus of this paper is on CEST, which aims to extract Z-spectral peaks from background signals (e.g., MTC and direct water saturation), I believe the ST metric shouldn't be used. In Figure 2, the regional contrast from amides and glycogen is very similar which indicates larger contributions from water properties and MTC. Can the authors quantify the separate signals using multiple-pool fitting or background curve fitting via R1rho theory? Additionally, for validation, can the authors compare their maps with other non MR techniques?

Author Response: Also see R1.2, 1.3, 1.5, 1.6, and the previous page of the revision summary. We have replaced the less-specific ST images in the main text with quantified CEST images, with all five contrast maps following the same standard all-through the manuscript (an overview table of human protocol and contrast definition displays in the next page, also in **Supplementary Table 1.1**). As suggested, **we have also added new Fig. 8 to directly compare SSE-CEST with ¹⁸F-FDG-PET/CT.** Multi-slice amide-CEST (LD, 3.5 ppm, 0.7 μ T) and glucose-CEST (MTRasym, 2.1 ppm, 2 μ T) images show good correspondence with the PET/CT-identified metabolic hot spots.

For the quantification, we really appreciate the reviewer's suggestions for adding the spectral-based analysis, which could further strength the full-spectral acquisition benefit of our SSE-CEST. We agree that the ST metric ($ST = 1 - Z$) is less specific, including contributions from water and MTC of macromolecules. Previously we mainly focused on describing the novelty in acquisition,

therefore mainly compared the raw ST images due to the limited space and the lack of a consensus of CEST quantification for multiple peaks under 3T.

As suggested by the reviewer, now we have performed two kinds of quantitative analysis based on background curve fitting, which are the Lorentzian Difference (LD) analysis and the R1 ρ -based method. Briefly, these two methods first estimate the background contaminations from water direct saturation (DS) and/or magnetization transfer contrast (MTC), and then take subtraction of the experimental measurements and the estimated backgrounds for CEST quantification. Under 3T field strength with line-broadening and peak overlap of CEST spectra, semi-quantitative methods using background fitting were often chosen, which fitted much less parameters and obtained better robustness than the multi-pool fitting (i.e., direct fitting the spectral peaks from the individual solute pool.) In the revised main text, we utilized LD methods for 0.7 μ T SSE-CEST protocol, in experiments of ex vivo porcine liver, glycogen phantoms, fasting volunteers and tumor patients, achieving three contrast maps including amide CEST, glycogen NOE and aliphatic NOE. While in the 2 μ T SSE-CEST protocol, we utilized asymmetric analysis and output APTw images and GlycanCEST images.

We also supplied the R1 ρ fitting for the fasting experiments (**Supplementary Fig. 5.5**), whereas for the OGTT and for the tumor patients, it was not applicable due to lack of T1 maps, T2 maps and the saturation B₁ maps. The contrast definition, the acquisition protocols and the quantification metrics are summarized below.

Supplementary Table 1.1. In vivo abdominal SSE-CEST protocols.

Contrast maps	Acquisition protocols	Quantification Metrics	Notes
Glycogen NOE (-1.2ppm)	Zneg (0.7 μ T): 10 parallel z- ω lines plus S ₀	LD analysis for fasting experiments and tumor patients; R1ρ analysis for fasting experiments where additional T1 and T2 maps were acquired.	Single-pool LD analysis were utilized for all 0.7 μ T data from SSE-CEST and from conventional CEST, with both good image quantity and spectral profile.
Aliphatic NOE (-3.5ppm)			
Amide CEST (3.5ppm)			
APT_w	Z (2 μ T): 11 parallel z- ω lines plus S ₀	MTR _{asym} (3.5ppm)	Recommended by 3T clinical consensus for glioma.
GlycanCEST (GlucocEST)		MTR _{asym} (1.2ppm)	Often used for faster hydroxyl exchange.

*OGTT and the comparison with PET used 2.1 ppm for 2 μ T protocol as suggested by the spectra.

The SSE-CEST acquisition patterns for B₁ of 0.7 μ T are displayed below (**Supplementary Fig. 5.1**), with the 10 parallel z- ω encodings plus S₀, i.e., Zneg acquired frequency offsets of -9, -6, -5, -4, -3, -2, -1, 0, 1.5, 3 ppm for the NOE contrast, and Zpos acquired frequency offsets of -9, 6, 5, 4, 3,

2, 1, 0, -1.5, -3 ppm for the CEST contrast. Noted that the 0.7 μT protocols sampled denser offsets when closing to water than the 2 μT protocol for capturing the line-shape of slower-exchanging amides, guanidium-amines and NOE effects. Notably, we employed a Dixon-type 3D fast gradient echo sequence with multi-echo readouts (3D-TFE), which directly output the water maps, the fat maps and the B0 maps (**Supplementary Fig. 4.1**). Our SSE-CEST reconstructed spectral dataset using the water maps, followed by a registration-free B0-correction using the Dixon B0 map.

Supplementary Figure 5.1. The trajectories of Zneg (0.7 μT) and Zpos (0.7 μT) for fasting experiments and tumor patients. **a**, the trajectories of Zneg (0.7 μT). **b**, the trajectories of Zpos (0.7 μT).

Supplementary Figure 4.1. Dixon-type 3D gradient-echo acquisition sequence for human abdominal SSE-CEST

The two spectral-based quantification methods are also introduced below, with examples in fasting volunteers displayed in **Supplementary Methods 2, 3** as well.

1) The Lorentzian-Difference (LD) analysis:

Under a relatively small saturation B_1 ($B_{1, \text{sat}} = 0.7 \mu\text{T}$ here), the contribution from direct saturation (DS) of water could be described as a Lorentzian line-shape (Zaiss, Schmitt et al. 2011). For

a small $B_{1,sat}$ (e.g. $0.7 \mu T$), MTC is also small and could be either neglected (Jones, C 2012 MRM)(Jones, Polders et al. 2012) (Dula, Arlinghaus et al. 2013), or considered as a constant when saturation frequencies are close to water (e.g. -10 ppm to 10 ppm)(Zaiss, Schmitt et al. 2011, Desmond, Moosvi et al. 2014, Deshmane, Zaiss et al. 2019). In the so-called LD analysis, the background Z spectra (Z_{CEST}^{LF}) is firstly fitted as reference without CEST contribution, and then CEST signals are extracted by subtracting the fitted background Z_{CEST}^{LF} and the acquired Z spectra (Z_{CEST}^{exp}).

$$LD = Z_{CEST}^{LF} - Z_{CEST}^{exp} \quad (\text{Eq.1})$$

Herein for a small B1 under 3T scanners, we fitted the Z_{CEST}^{LF} by a single-pool Lorentzian function for DS, with the contribution of MTC considered as a constant baseline:

$$Z_{CEST}^{LF} = 1 - L_{DS} - MTR_c = b - \frac{A \cdot \Gamma^2 / 4}{\Gamma^2 / 4 + \Delta\omega^2}, \quad (\text{Eq.2})$$

in which A denotes the magnitude of water line, Γ is the line-width, $\Delta\omega$ is the frequency offset from water, and b is a baseline. Quantification based on single-pool LD methods were employed in quantifying human muscle glycogen (Bie, Bo et al. 2025), and was also preliminarily validated on human fasting experiments using a free-breathing liver imaging sequence (Xu, Leforestier et al. 2025), both showing good robustness. To fit in the single-breathing time scale, our SSE-CEST separately acquired the positive part ($Z_{pos_0.7 \mu T}$) and the negative part of Z spectra ($Z_{neg_0.7 \mu T}$), both using a saturation B1 of $0.7 \mu T$ same as previous study (Xu, Leforestier et al. 2025).

Taking the fasting experiments as an example, we displayed the SSE-CEST Z spectra (Z_{lab} , in blue), the fitted background (Z_{LF} , dash line in black) and the subtracted one (LD, in magenta, pinkish-purple). The detailed LD spectral analysis was displayed below (**Supplementary Figs. 5.2-5.3, for Z_{pos} fitting and Z_{neg} fitting**), for the subject in revised **Fig. 5**.

Supplementary Figure 5.2. Lorentzian Difference analysis of Z_{pos} ($0.7 \mu\text{T}$) for the fasting experiments. Upper Row: 2hr-post meal, Lower Row: fasting, **a,c** for a liver ROI and **b** for a muscle ROI. As seen, compared with liver, muscle has a stronger MTC ($>20\%$), while liver obtains a higher CEST spectra.

Similarly, the NOE signals from glycogen (-1.2ppm) and from aliphatic protons (-3.5ppm) could be quantified using LD, for the Z_{neg} ($0.7 \mu\text{T}$) acquired within a single breath-hold (Zhou, van Zijl et al. 2020, Bie, Bo et al. 2025).

Supplementary Figure 5.3. Lorentzian Difference analysis of Zneg (0.7 μ T) for extracting NOE signal in the fasting experiments. Upper Row: 2hr-post meal, Lower Row: fasting, **a,c** for a liver ROI and **b** for a muscle ROI. Glycogen NOE peaks (\sim 1.2 ppm) were observed on all 4 subplots. While liver exhibited dropped glycogen NOE signals after overnight fasting (Also see **revised Fig. 5**). Muscle has a stronger MTC than liver has.

Supplementary Figure 5.4. Spectral comparison between 2hr-post meal and over-night fasting. **a,b**, LD analysis for fasting (dash) and post-meal (solid), with blue for liver, black for muscle. **c,d**, the spectral difference between fasting and meal (solid: Δ ST spectra, dash: LD spectra), with magenta for liver, black for muscle.

In **Supplementary Fig. 5.4**, we put together the LD spectra acquired 2hrs-post meal (m.) and those post overnight fasting (f.). As seen, the m. liver exhibited clear spectral peaks at amide offsets (~3.5 ppm) and at the glycogen NOE frequencies (-1 ppm to -2 ppm). Whereas the 1/3 reduction in LD(1.2 ppm) post overnight fasting was consistent with reported in a very recent study (15 min scans with intensive registration required, (Xu, Leforestier et al. 2025)). When taking the difference of the raw Z spectra and the LD at two time points, ΔST of liver indeed include a 5%-10% baseline contribution (the flat part from 6 to 8 ppm, -6 to -8 ppm), making the frequency-specific CEST or NOE effect not clear. In contrast, the pinkish-purple LD line well denotes the amide peak, the glycogen NOE and the aliphatic NOE peak.

Noted that for several liver voxels, the edge of Z-spectra may experience ‘rising tail’ (**Supplementary Figs. 5.2, 5.3a,c**). This is presumably because, liver tissues are rich in blood with long T1 and flow effect, therefore the system did not read steady-state when the edge freq. was scanned at the beginning. Also noted that, due to the Gradient-echo readouts and a relative-high noise level using single-shot, the lowest value of Z spectra at water frequency always not zero (Deshmane, Zaiss et al. 2019). Furthermore, our data suggested liver has a higher value than muscle when $\Delta\omega = 0$, this could due to the substantial blood volume and fast blood flow in liver. This may cause a negative value for glycogen NOE (used LD (-1.2ppm) here). For statistics, we added constrains to the amplitude boundary (1 to $+\infty$) and the line-width (0 to 1.8) for the single-pool Lorentzian fitting, while the values at -1.2ppm subtract the values at -7 ppm (the common far-most offset for all slices) for quantification.

To deal with the above imperfection of in-vivo single-breath-hold abdominal imaging, we improved the LD analysis by integrated a QC pre-correction step as below:

1. **Trimmed the dropped part** at the Z spectral edges, eliminating the data that did not reach steady-state, non-perfect B0 correction or have cardiac motions;
2. **extrapolate the 9.5 ppm, 10 ppm points of Z spectra**, using the value of MTC effect by averaging MTR values between ± 6 ppm and $\Delta\omega_{max, slice}$ (the far most offset of the slice).

Revised Fig. 5a,b. SSE-CEST (0.7 uT) enabled extraction of three contrast maps using LD analysis, including amide CEST, glycogen NOE and aliphatic NOE.

Since the LD analysis does not require T1 or T2 mapping data, we could **utilize the same analysis**

codes for both healthy subjects and tumor patients. The represented LD spectra from tumor patients are displayed in **Supplementary Fig. 7.2**, which obtained better contrast interpretation and quality control capability than a single contrast map.

2) The R1ρ-based background fitting:

Alternatively, we also employed analysis based on R1ρ physical model, rather than the LD analysis base on spectral line-shape. Briefly, R1ρ describes the decay rate of longitudinal magnetization in a CEST or Spin-Lock MR experiment (Jin, Wang et al. 2012, Zaiss and Bachert 2013). The metabolic-specific contributions, termed as exchange-dependent relaxation in the rotating frame (Rex), could be extracted by removing the R1ρ effect from background water (R1ρ,w):

$$R_{ex} = R_{1\rho} - R_{1\rho,w} = R_{1\rho} - (R_{1w} \cos^2\theta + R_{2w} \sin^2\theta) \quad (\text{Eq.3})$$

Where $\theta = \text{atan}\left(\frac{\gamma B_1}{\Delta\omega}\right)$, for a CW-type saturation pulse employed in our SSE-CEST. The measured

R1ρ, could be calculated from the acquired Zspectra, by

$$Z = \left(\frac{1 - e^{-R_{1a}TR}}{1 - e^{-R_{1a}TR} \cos(\alpha)} \cos(\alpha) - M_{SS} \right) e^{-R_{1\rho}T_S + M_{SS}} \quad (\text{Eq.4})$$

$$\text{in which } M_{SS} = \frac{R_{1a}}{R_{1\rho}} \cos^2(\theta)$$

Noted that in Eq.4, we implemented the effect of small-flip-angle(α) gradient-echo readouts that SSE utilized.

R1ρ quantification was demonstrated in the fasting experiments, where the endogenous T2 relaxation could also be changed due to a reduction of glycogen (Yadav, Xu et al. 2014, Shizhen Chen 2023). For the same subject in revised Fig.5 (0.7uT), we extracted Rex contrast maps according to Eq.3 and Eq.4, with the contrast maps and the Rex spectral profile added in **Supplementary Figure 5.5** (next page). Similar to the LD maps, the amide CEST and glycogen NOE maps also displayed significant drops post over-night fasting. Since a single-slice T2 mapping was acquired for 4 subjects in the fasting experiments, we also compared the T2 values at 2-hour post meal and post overnight fasting, which indeed also displayed an increase of T2 values post overnight fasting presumable due to the lower content of glycans (**Supplementary Figure 5.5c**). The R1ex spectra also displayed a gap at amide frequency (~3.5ppm), the hydroxyl CEST frequency (~1 ppm) and at the glycogen NOE offsets (0.5ppm-2ppm). However, since R1ρ-based quantification required acquisition of T1 and T2 mapping sequences, which was not available in our current protocol for patient data. Therefore, in revision we only retrospectively analyzed the existed SSE-CEST data from liver tumor patients, using the LD analysis for the 0.7 μ T SSE-CEST, and using asymmetric MTR (APT_w and GlycanCEST) for the 2 μT SSE-CEST respectively.

Supplementary Figure 5.5. R1 ρ -based quantitative analysis of metabolic and relaxation changes before and after overnight fasting in healthy volunteers. a,b, Representative maps from one subject showing T₁, T₂, and the exchange-dependent relaxation rate (R_{ex}) for three specific contrasts—amide CEST (3.5 ppm), glycogen NOE (-1.2 ppm) and aliphatic NOE (-3.5 ppm)—acquired **a**, two hours after a meal and **b**, after a 12-hour fasting. R_{ex} values are notably reduced after fasting, while T₁ and T₂ maps show no appreciable change. **c**, Group-wise comparison of T₂ values (n = 4 subjects). Although a statistically significant decrease is observed post-meal (p < 0.001), the absolute difference is small (mean \pm SD: 77.2 \pm 11.2 ms post-meal vs. 81.9 \pm 11.9 ms post-fasting), indicating that water relaxation changes, while detectable, do not dominate the observed metabolic contrasts. **d**, R_{ex} spectra of liver parenchyma, demonstrating a consistent reduction in the exchange-related signal after fasting.

Theoretically, owing to the fast acquisition of full-spectra from our SSE-CEST, several quantification methods may also be applicable in future especially under multi-B1 acquisitions, including PLOF (Chen, Barker et al. 2019) and multi-pool Lorentzian fitting combined with deep-learning (Glang, Deshmane et al. 2020). Overall, the full-spectral, motion-stabilizing acquisition of SSE-CEST will improve the interpretation of endogenous CEST signals *in vivo*.

To validate metabolic specificity, we compared SSE-CEST with 18F-FDG PET/CT in an HCC patient showing a PET hot spot 10 months post-treatment (Fig. 8). SSE-CEST contrast maps acquired 3-week after PET scan revealed matched hyper-intense regions (red arrows) on three representative axial slices. The SSE-CEST maps included the amide-specific LD contrast at 3.5 ppm acquired with a low saturation power (B₁ = 0.7 μ T), reflecting mobile protein/peptide content, and the glucose-

weighted ST contrast at 2.1 ppm acquired with a higher power ($B_1 = 2 \mu\text{T}$). Both contrasts also display adjacent necrotic regions (white arrows). This agreement with PET/CT and Gd-enhance MRI suggest the potential of SSE-CEST for ‘label-free’ non-radiative way in monitoring treatment response.

Figure 8. Comparison of SSE-CEST metabolic maps and FDG-PET in a patient with hepatocellular carcinoma (HCC), scanned post treatment. Three axial slices are shown, highlighting active HCC lesion (red arrows) and adjacent post-treatment necrotic regions (white arrows). **a**, anatomical T1 weighted image. **b**, SSE-CEST metabolic maps: (left) amide CEST (LD (3.5 ppm)) using B_1 of $0.7 \mu\text{T}$, reflecting mobile protein/peptide content; (right) the ST image at 2.1 ppm (B_1 of $2 \mu\text{T}$), reflecting glucose-related metabolism. **c**, Corresponding slices on the fused images of ^{18}F FDG-PET and CT. The hyper-intense regions on SSE-CEST images matched well with the ‘hot-spot’ regions on PET images, which indicates higher glucose metabolism.

Author Actions:

1. Updated **all the ST images in main text with extracted CEST maps**, following the standardized quantification protocols in **Supplementary Table 1** (except for **Fig. 4** for acquisition repeatability and three orthogonal visualization);
2. Validated the SSE-CEST quantification using two kinds of background fitting, LD and R1p methods, in the fasting experiments (**Supplementary Methods, Supplementary Figs. 5.1-5.5**);
3. Strengthened the benefits from SSE-CEST’s full-spectral and motion-stabilizing acquisition, including better quantification, contrast interpretation, and quality control (**discussion and results**);

4. Added new **Fig. 8** to directly compare SSE-CEST with ^{18}F -FDG-PET/CT.

[Comment 1.2]

2. In the difference spectra shown in Figure 2c, f, i, it is clear that the z-omega pattern does affect Z-spectral features. The Bland-Altman plots shown in Figure S7 are comparing whole Z-spectra which are primarily determined by water properties and MTC. CEST signals tend to be much smaller (approx 1-3%). It would be more important to compare extracted CEST signals with the different sampling schemes.

Author Response: Thanks for pointing this out. See author actions in R1.1, we have updated the previous RAW ST images in Fig. 2 using the quantified CEST images, including amide (LD (3.5 ppm)), glycogen NOE (LD (-1.2 ppm)), and aliphatic NOE (LD (-3.5 ppm)). Noted the definition of contrast maps are consistent all through the manuscript. For three SSE-CEST acquisition patterns, we compared both its Z-spectra and the extracted LD spectra with those acquired from conventional CEST (See below the revised Fig. 2).

Figure 2. Ex vivo porcine liver metabolic imaging with SSE-CEST using variable z- ω acquisition patterns.

Parallel (a1-c1), diamond (a2-c2), and radial (a3-c3) SSE-CEST are compared with conventional CEST (a4-b4). The left column (a1-a4) shows the respective acquisition patterns in z- ω plane. The middle columns (b1-b4) present multi-contrast maps (Amide CEST, Glycogen NOE, and Aliphatic NOE) from a representative axial slice, with their

locations in z - ω plane marked by black arrows in **a1-a4**. Bland-Altman analysis (the bottom plots in **b1-b3**, $n = 69$, the number of liver ROIs (8×8 square) within the displayed slice) demonstrates good agreement between each SSE-CEST pattern and conventional CEST. Mean difference in solid lines and 95% Limits of Agreement in dashed lines (see quantitative results in **Supplementary Table 1.2**). The right columns (**c1-c3**) display CEST spectra (Z and LD spectra) from the region of interest indicated by a red open box in the T1-weighted anatomy (**d**). Spectra are color-coded (SSE-parallel: blue, SSE-diamond: magenta, SSE-radial: red; dashed lines for SSE-CEST, solid gray lines for conventional CEST). The near-zero difference spectra (solid lines around zero) confirm minimal deviation between SSE-CEST and conventional CEST. All data were acquired with $B_1 = 0.7 \mu\text{T}$ and quantified using LD. Further statistics and comparison across all slices are in **Supplementary Table 1.2** and **Supplementary Fig. 2.1-2.3**.

As seen, for the displayed slice (pointed by the black arrows), three SSE acquisition patterns almost have identical Z spectra as the conventional CEST, for a ROI displayed in **revised Fig. 2d** (**revised Fig. 2c1-c3**). Yet for the extracted LD spectra, there are indeed slight mismatches when $\Delta\omega$ getting closer to water. But also noted that three patterns acquired distinct numbers of z - ω encoding lines (i.e. acquisition time), with 30 lines for the ‘parallel’ and ‘diamond’ pattern (~ 4 min. 15 sec.) and 18 lines for ‘radial’ pattern (~ 2 min. 50 sec.).

This experiment successfully demonstrated the flexibility of SSE-CEST, achieving high-fidelity spatial-spectral reconstruction. Specifically, contrast images exhibited well-preserved anatomical details (e.g., major vessels), while spectra maintained excellent quantitative accuracy. Furthermore, these findings indicate that SSE-CEST acquisition patterns can be prospectively customized and optimized, according to application-specific requirements, including target metabolite frequencies and anatomical regions of interest (**added to discussion**).

Author Actions:

1. Updated the multi-metabolite ST images (**Fig. 2, middle columns**) with the quantified CEST images (extracted using LD analysis, see **revised Fig. 2 b1-b4**). Now all the quantitative images were consistent although the manuscript: Amide CEST (3.5ppm, LD@0.7uT), glycogen NOE (-1.2 ppm, LD@0.7uT), and aliphatic NOE (-3.5 ppm, LD@0.7uT);
2. Added the comparison of quantitative LD spectra, as suggested by the reviewer. The revised **Fig. 2c1-c3** presents both Z spectra and extracted LD spectra from SSE-CEST, demonstrating that all three acquisition patterns preserve the spatial-spectral information of the conventional CEST (gold standard);
3. Improved the writing of results and figure captions, clarifying the pros and cons for each trajectory;
4. Added in the discussion the flexibility of z - ω trajectory design (also see **Comments 1.3**).

[Comment 1.3]

3. In Figure 2, it was determined that a parallel trajectory was the most reproducible. Can the authors explain why this was? Is it possible that it is due to only the frequency offset being changed with no changes in the gradient?

Author Response: We thank the reviewer for the insightful comments. We attribute the superior reproducibility of the parallel trajectory into three factors:

- 1) **Simplified and consistent encoding: We agree with the reviewer's intuition.** The parallel trajectory only varies the frequency offset ($\Delta\omega$) among all acquisition z- ω lines, with a constant saturation gradient (G_{sat}). By excluding other measurement variations, this ensures that the fundamental spatial-spectral encoding relationship ($\Delta\omega = \gamma \cdot G_{\text{sat}} \cdot \Delta Z$) remains identical for every encoding step. Consequently, **all slices along the z-direction experience the same frequency step size and, crucially, the same saturation bandwidth** (the inherent frequency spread per slice induced by G_{sat}). This consistency minimizes a potential source of error during acquisition and reconstruction.
- 2) **Non-Overlapping sampling:** The parallel lines in z- ω space do not intersect, ensuring that each sampled point is unique. This provides the most efficient coverage of the z- ω plane for a given number of encoding steps and avoids potential ambiguities that could arise from overlapping sampling points in other trajectories (e.g., diamond, radial). E.g. for the displayed central slice (S21), diamond pattern had half of sampling ω overlapped, making the total acquired ω cut by half. This may cause the reduced accuracy in reconstruction, especially for frequencies closer to water (Glycogen NOE image, -1.2ppm here).
- 3) **equal ω intervals for each slice:** We adopted a parallel acquisition with equal ω intervals, leading to the same sampling density of ω for all slices. This could reduce the reconstruction and quantification bias due to varied sampling intervals among slices.

In summary, based on above considerations, we chose parallel acquisition pattern in this very 1st technical paper for single-breath-hold liver CEST imaging.

Nevertheless, all three trajectories of SSE-CEST did NOT show significant difference with the conventional CEST as gold standard (G.S.), in amide CEST, glycogen NOE and aliphatic NOE signals (**Supplement Fig. 2.1**). All showed similar contrast with the conventional CEST, **enabling flexible trajectory design tailored to specific application scenarios**, subject to further optimization according to targeted metabolites and organs of interest.

Supplementary Figure 2.1. Quantitative comparison of CEST contrast maps between SSE-CEST and conventional CEST in ex vivo porcine liver. (a) Amide CEST, (b) Glycogen NOE, and (c) Aliphatic NOE. Results from SSE-CEST with parallel, diamond, and radial z - ω acquisition patterns are displayed. All data were acquired with $B_1 = 0.7 \mu\text{T}$ and quantified using LD. Data are presented as mean \pm SD ($n = 69$, the number of liver ROIs (8X8 square) within the displayed slice). **ns** represents $p > 0.05$ and ******** represents $p < 0.0001$.

As seen in **Supplementary Figure 2.1**, only the aliphatic NOE acquired from radial trajectory displayed statistical difference with the G.S.. This mismatch maybe because radial pattern acquired only 3/5 of z - ω lines compared with the other two SSE patterns. As seen in **Fig. 2a3**, the radial acquisition only sampled very few points in the aliphatic NOE range, therefore for the chosen slices resonating closer to water, it may miss the ‘labeling’ ω or result in inaccurate B_0 correction. This highlights a significant avenue for future optimization: the design of advanced z - ω trajectories that can further reduce the number of encodings while maintaining high fidelity, thereby pushing the boundaries of acquisition efficiency for clinical applications where scan time is paramount. For the current study, which prioritized robust and reproducible metabolic mapping, the parallel trajectory was the optimal choice. In contrast, trajectories that vary G_{sat} introduce an inconsistency: the saturation bandwidth becomes a variable across encoding steps, which can complicate the spectral profile, as we noted in the Discussion.

Supplementary Figure 2.2. Overview of slice variation for SSE-CEST with parallel z- ω acquisition pattern in ex vivo porcine liver. (a) Amide CEST, (b) Glycogen NOE, and (c) Aliphatic NOE. A total of 21 slices are shown. All data were acquired with B1 = 0.7 μ T and quantified using ST.

We have also compared the variations among slices for SSE-parallel and for conventional CEST, using Bland-Altman plots, as displayed below.

Supplementary Figure 2.3. Bland-Altman analysis between the mean signal on all 21 slices acquired by SSE-

CEST and by conventional CEST (parallel z- ω acquisition pattern 0.7 μ T, ex vivo porcine liver). (a) Amide CEST, (b) Glycogen NOE, and (c) Aliphatic NOE. Each circle denotes the average value of a ROI as denote in Fig. 2d (n = 21), one ROI per slice across 21 slices. Mean difference in solid lines, with the 95% Limits of Agreement in dashed lines indicating the small variance among slices.

Author Actions: Add to discussion. Also see R1.2, revised Supplementary Figure 2.1

[Comment 1.4]

4. In Figure 3, and similar to before, I think it is incorrect to use the ST metric to characterize the glycan and glycogen signals. Background fitting or multiple-line fitting should be used.

Author Response: We agree with the reviewer and have replaced ST metric with the more quantified metrics all through the text (See similar comments R1.1, 1.2 1.5 and R2.3).

For glycogen NOE signal, SSE-CEST was acquired using B1 of 0.7 μ T, with LD analysis for quantification (fitting background DS and considered MTC as a constant offset).

Author Actions:

According to comments R1.5, R2.3, we have re-performed the phantom studies, with agar and Gd-agents added for mimicking liver tissue. The ST metric have also been replaced with more quantitative analysis. See revised Fig. 3 and R1.5.

[Comment 1.5]

5. What is causing the variation in contrast across the slices and within the tubes in Figure 3? I am assuming the mixture is homogeneous.

Author Response: Previously glycogen phantom was prepared in free water (PBS) and the tubes were laid on horizontally along the magnet direction. Therefore, we assume the contrast variation could come from the motion of free water, whereas the long T1 and T2 may also cause spectral oscillations. In the revised version, we have re-performed phantom experiments, by adding 1% agar and getting crosslinked. The T1 have also be titrated using Magnevist (a clinical-approved Gd-agent). Now for the new phantom the fluid motion was removed, with similar relaxation and MTC effect with tissue. The contrast across the slices shows consistency in the updated Figure 3. (Also see R2.2)

revised Figure 3. Quantitative evaluations of SSE-CEST using glycogen phantoms in 1% agar. **a**, Phantom layout in a x-y slice. **b**, Illustration of different slices along z direction. **c**, Glycogen NOE (LD at -1.2 ppm) maps for slice #9, 11, 13, displaying a linear-correlation with the glycogen concentration (**d**, $n = 21$ slices, $\text{mean} \pm \text{std.}$) **e**. Z spectra from SSE-CEST for each tube in slice #11, with the symbol 'x' marked the actual acquisition points. The specific frequency range from glycogen NOE are zoomed in the right bottom (-0.5 ppm to -2 ppm). **f. h.** Scattering plots and Bland-Altman plots of glycogen NOE values, comparing between SSE-CEST and conventional CEST (both with $n = 126$, 6 tubes X 21 slices). **g.** CEST LD spectra derived from **d**, highlighting cleaner NOE signal peaks (-0.5 to -2 ppm) and hydroxyl signals (0.5 to 2 ppm). $B_1 = 0.7 \mu\text{T}$ and parallel z- ω pattern were used for this figure. Results for other acquisition parameters are displayed in **Supplement Fig.3.1,3.3-3.4**, including slice-by-slice image comparison and statistical comparison with fully-sampled conventional CEST, and quantitative correlation with glycogen concentration.

Supplementary Figure 3.4. The Z-spectra of three slices for the 150 mM glycogen tubes in main Fig. 3, with three types of markers denoting their distinct sampled ω . As seen, the reconstructed spectra are almost identical, with the slight difference due to the inhomogeneity when crosslinked with 1% agar. Also see the slice-by-slice comparison with conventional CEST in Supplementary Figs. 3.2-3.3.

Supplementary Figure 3.2. Slice-by-slice comparison ST (-1.2 ppm) images in glycogen phantoms in 1% agar,

acquired using SSE-CEST and using conventional CEST, both with a B1 of 2 μ T.

Supplementary Figure 3.3. Slice-by-slice comparison of ST (1.2ppm) images in glycogen phantoms in 1% agar, acquired using SSE-CEST and using conventional CEST, both with a B1 of 2 μ T.

The slice-by-slice comparison suggested that SSE-CEST images and full-sampled conventional exhibited almost identical images, with no statistical difference observed (n=126, 21 slices each with 6 tubes, $p < 0.0001$, **Supplementary Fig. 3.1g,h**). Additionally, both ST (1.2 ppm) and ST (-1.2 ppm) are proportional to Glycogen concentrations (**Supplementary Fig. 3.1e,f**), which are attributed to hydroxyl CEST signals on glycans (1.2 ppm) and the relayed NOE signals from macromolecular glycogens (-1.2 ppm), respectively. It demonstrated the feasibility of using 2 μ T with a shorter saturation time for in vivo detection of slower exchange.

Supplementary Figure 3.1. Glycogen phantom experiments: same layout as Fig. 3 but with saturation B1 2 μ T. **a**, phantom layout in x-y slices. **b**, illustration of different slices along z direction. **c**, ST (1.2 ppm) images for ‘labeling’ hydroxyl CEST on glycans and ST (-1.2 ppm) images for ‘labeling’ the replayed NOE from macromolecular glycogen respectively; with slices# 3, 6, 9, displayed. **d**, the Z-spectra of each tube on slice 11, illustrating the featured frequency range for glycan-CEST and glycogen-rNOE; **e**, **f**, concentration dependence of ST(1.2ppm) and ST(-1.2ppm) from SSE-CEST (n=21 slices, mean \pm std) **g**,**h**, the scattering plot of glycan-CEST(**h**) and glycogen-rNOE (**g**) values, comparison between conventional methods and the parallel-encoded SSE methods. Each point indicates a tube ROI (6 ROIs \times 21 slices).

Author Actions:

1. Reperformed the phantom experiments, by dissolving glycogen in 1.5% crosslinked agar instead of PBS. Therefore, motion of free-water was reduced and the T1, T2 values are closer to tissue. (Revised Fig.3, Supplementary Fig.3.1-3.5)
2. Added a slice-by-slice supplementary figure, for both ex vivo porcine liver and phantom tubes, visually and statistically compared the variations among slices. (Supplementary Fig.2.1-2.3, 3.1-3.5)
3. Added the slice-by-slice comparison of ST images from SSE-CEST and from the conventional one, which further validate the high-fidelity reconstruction, quantitation and the reliability of SSE-CEST. (Supplementary Fig.3.3,3.4)

[Comment 1.6]

5. The methods section states that human experiments were performed using either 2 uT or 0.7 uT. The 2uT results shown in Figures 4 and 5 show that almost 90% of the signal was saturated at 1.2/-1.2 ppm. What did the 0.7 uT results look like? Did they have higher SNR? Also, it is unclear what is being reported on at these frequencies.

Author Response: Also See R1.1, R1.8. As reviewer suggested, we have added 0.7 μ T data in revised Fig. 5, 7, by displaying five kinds of standard contrast maps (Supplementary Table 1), including amide CEST (Zpos_0.7 μ T LD(3.5ppm)), glycogen NOE (Zneg_0.7 μ T LD(1.2ppm)), aliphatic NOE (Zneg_0.7 μ T LD(-3.5ppm)), APTw (2 μ T MTRasym(3.5ppm)) and Glycan CEST (2 μ T MTRasym(1.2ppm)). Since 0.7 μ T allows the spectral-based quantitative analysis, which could further strength the value of our SSE-CEST, i.e., high-efficient acquisition of both spectral and spatial information.

For comparison, we also put the ST images acquired under 2 μ T and that under 0.7 μ T in the same figure, for the subject in the fasting experiments **in revised Fig. 5**. The SNR of SSE-CEST Z-spectral images are also compared under 5 ‘labeled’ saturation offsets. **We also would like clarify that the 80-90% signal loss at 1.2ppm/-1.2ppm images partially related with the acquisition order of reference S_0 .** For the in vivo protocol, we first acquired the reference z- ω line centered at -1560 ppm as S_0 , followed by the 10 or 11 parallel lines. Therefore, the starting magnetization of S_0 could be larger than the following saturated ones, resulting in an apparent larger value (up to 90%) due to the ‘exaggerated’ denominator. **But as below, the quality of ST images (1-Z) and the SNR values of SSE-CEST raw Z-spectral images all look good. (Also See R2.X)**

Supplementary Figure 5.6. For a subject in the fasting experiment, comparison of SSE-CEST images and the SNR values acquired using $0.7 \mu\text{T}$ and those using $2 \mu\text{T}$. **a.** ST images (ST=1-Z) of $0.7 \mu\text{T}$ acquired 2-hrs post-meal and post overnight fasting, at 5 distinct frequency offsets, corresponding to the labeling of Glycogen NOE (-1.2 ppm), -OH on glycans (1.2 ppm), guanidine amines (2 ppm, also includes -OH leakage under $2 \mu\text{T}$), amide (3.5 ppm)

and aliphatic NOE (-3.5 ppm). **b.** same layouts as **a**, but using B_1 of 2 μ T. **c.** SNR comparison of Z spectral images acquired using 0.7 μ T and 2 μ T, at 2 hours post-meal. The SNR values were calculated 10 times, using a randomly-selected noise region (7×7 square) in the background and the signal from the same central liver region (SNR = mean(Signal)/std(Noise), N=10). **d.** the same layouts as **c**, but acquired post over-night fasting. For the frequency offsets of **-1.2 ppm** and **3.5 ppm**, the SNR of 0.7 μ T are significantly higher than those of 2 μ T (*** $p < 0.005$, ** $p < 0.01$, * $p < 0.05$).

Author Actions:

1. **Add three types of quantitative contrast maps and the spectra acquired under 0.7 μ T, all through the main text and the Supplementary** (Except for Fig. 4 with repeatability evaluations under 2 μ T).
2. Add comparison of the SNR and the raw ST images acquired under 0.7 μ T and under 2 μ T, in the supplements (**Supplementary Figure 5.5**).

[Comment 1.7]

7. It is sometimes difficult to follow the signals reported and what they are meant to represent. For instance, for healthy volunteers, amides are at 3.5 ppm, amines from 2-3 ppm, and hydroxyls from 0.6 to 2 ppm. In the subsequent results involving fasting, signals are reported at 3.5, 2, 1.2, -1.2, and -3.5 ppm. Is 2 ppm supposed to be amines or hydroxyls? In the glucose ingestion experiments, MTRasym is reported in the main document and also linewidth in the supplementary. The parameters keep on changing throughout the paper and are hard to follow. I would like to see a consistent parameter used and one that is as specific to the molecular origin of the contrast.

Author Response: Thank you very much for pointing this out. In the revised version, **we have made all the contrast maps consistent throughout the manuscript**, with the ST images and extracted CEST maps clearly marked.

Supplement Table 1 overviewed the definition of three SSE-CEST protocols (Zpos_0.7 μ T, Zneg_0.7 μ T and Z_2 μ T) and five kinds of contrast map (three from LD analysis of the 0.7 μ T protocol and two MTRasym maps from the 2 μ T protocol).

Regarding to the 2 ppm signals, we now remove it from the endogenous CEST imaging, due to the lack of specificity. Indeed, when applying a saturation pulse at 2 ppm (2 μ T), several kinds of metabolites could be labeled including creatine, guanidinium amines on mobile proteins, and spectral leakage from glycanCEST (hydroxyls). However, for the exogenous glucose imaging, we chosen MTRasym (2.1ppm) according to the spectral changes. **Also see R1.9.**

Author Actions:

1. Consistent definition of contrast maps all through the text, added **Supplementary Table 1**.
2. The 2 ppm ST signals from endogenous imaging was removed due to lack of specificity.

[Comment 1.8]

8. In Figure 5, there is a statistically significant signal change for ST(3.5ppm) bit not APTw. Does this

indicate the changes are mostly due to changing water parameters?

Author Response:

Also see R1.1, R1.6,1.7.

For 4 out of the five subjects in the fasting experiments, we acquired T2 maps for a middle axial slice, which indeed suggested an increase of T2 post over-night fasting since glycans also exhibit T2-exchanging effect (Yadav, Xu et al. 2014) (**Supplementary Figure 5.5c, R1.1 in Rex plotting, the original figures are displayed below**). In comparison, the T1 maps of sub004 and sub005 did NOT changes much. We further performed R1 ρ -based analysis which eliminated the water relaxation changes (**Supplementary Methods 1.2**), and the resulted R1ex spectra suggesting a signal drop in the offset range of glycogen NOE (-0.5 to -2 ppm). (**Supplementary Figure 5.5**)

Response Fig.1 The single-slice T2 maps from four out of five subjects and T1 maps from two of them. (The statistical comparison of T2 post fasting and the resulted R1ex maps and spectra in **Supplementary Fig. 5.5**)

Previous **Fig. 5** displayed ST(3.5ppm) and the derived APTw maps acquired under B1 of 2 μ T, with the statistics performed by averaging signals from the entire liver (n=105, 5 subjects each with 21 slices). We attributed the lack of significant changes in APTw images to three main reasons.

1. Previous ROI plots of entire liver include upper left regions affected by cardiac motions, major blood vessels and regions with B1 inhomogeneity. This could affect APTw statistics.
2. APTw is a composite metric sensitive to noise, as it resulted from the subtraction of ST(3.5 ppm) and ST(-3.5 ppm). A concurrent reduction in both the amide and rNOE signals post-fasting could lead to partial cancellation in the APTw value (-5% to 5% range), masking changes in the individual components (**See R1.6**).
3. The ST (3.5 ppm) signal had the statistical difference post-fasting since it also includes the changes in water T2 (**Supplementary Fig.5.5**) and in the MTC, apart from amide CEST. As seen, the post-fasting MTC suggested a 5%-10% changes for 0.7uT (Δ ST(\pm 8ppm) as in **Supplementary Fig.5.5**).

To address this, we refined our statistical approach in the revised analysis. We increased the sample size and specificity by placing 3 small ROIs on the right liver (avoid left portion that easily affected by cardiac motion), resulting in 315 measurements (3 ROIs/slice \times 21 slices \times 5 subjects). As

seen in the revised Fig.5d, the APTw difference between post-meal and fasting states now reaches statistical significance (**p < 0.01). However, its effect size remains smaller than the Glycogen NOE acquired under 0.7uT (LD(-1.2 ppm), ****p<0.0001), consistent with its inherent composite nature.

Nevertheless, although ST images at 2 μT was not as specific as the LD analysis. The saturation ‘labeling’ is real and the ST signals displayed linear correlation in the glycogen phantom in agar (Supplementary Fig.2.1, both at the hydroxyl CEST (1.2ppm) and glycogen NOE(-1.2ppm)).

For *in vivo*, it costs less saturation labeling time (0.4 seconds versus 0.8 seconds for 0.7uT) and provides motion-stabilizing images with good tissue-contrast and easily-recognizable anatomy (as in the patient data, revised Fig.7,8 and supplementary Fig.7). As seen in Supplementary Fig.5.7 (previous Supplementary 5), the metabolic profile of multiple organs could be displayed. While the difference for the muscle spectra are close to zero for all frequencies, suggesting that muscle could be an internal reference for 2uT images.

Supplementary Fig. 5.7 (previous Supplementary 5) SSE-CEST imaging of multiple abdominal organs. a, ST image at 3.5 ppm acquired post fasting, with liver, kidneys, pancreas, muscle, spinal disk, and spleen sketched. b, CEST spectra were plotted for liver, pancreas, spleen and muscle, respectively. solid lines: post meal; dash lines: after overnight fasten; c, The subtracted spectra of pre- and post-fasting, indicating the metabolic characteristics for four different organs. The vertical stripes show the location of feature freq. offsets.

Author Actions:

1. Added **Supplementary Figure 5.4** for relaxation comparison and R1pho-based analysis for eliminating water background.
3. Improve sample size and the ROI plots by avoiding the upper-left regions affected by cardiac motions (**Supplementary Figure 5.1-5.7**)

[Comment 1.9]

9. Comparing the results in Figure 6, and Part 7 of the supplementary, it is not clear if the time course in Figure 6b is reproducible across the five subjects. I suspect the SNR is insufficient to measure changes due to glucose ingestion. In addition, given the relatively slow time course of glucose signal changes, is this rapid imaging technique suitable for this application?

Author Response:

We will respond this comment in two aspects, the improvement of the time-course plots and the

rational of using SSE-CEST in OGTT experiments.

For the time-course plots, we agree that previous plots in Supplementary 7 did not clearly reflecting the rising trends of glucose content within 1hr post oral-glucose take-up. i.e. the three kinds of metrics, including line-width, ST(2 ppm) and MTR_{asym} (2 ppm), displayed relative flat trends with small changes, lacking constancy across the five subjects. **We attributed this to two possible reasons**, one is the inclusion of voxels in upper-left liver which are affected by cardiac motion, resulting in large error bars among all the liver voxels. Another reason was the improper calculation of line-width using 2 μ T data, whereas it should be calculated using a smaller B₁ as in the dynamic glucose enhancement paper (Ref 31 in the main text). In the revised version, to ensure the robustness of the dynamic glucose CEST signal, we implemented a voxel-wise quality control procedure. Briefly, after manually delineating the entire liver parenchyma, the MTR_{asym} time course of every voxel was inspected. Voxels showing abrupt signal jumps (i.e., a change >0.05 within a 3-min interval, such as from 1% to 6%) were excluded as outliers likely arising from motion or local susceptibility artifacts. The final subject-averaged curves were generated using only the remaining stable voxels, thereby improving the reliability of the reported glucose dynamics.

Use the last version of Figure 6. We have added the Zspectral and MTR_{asym} spectral at two time-points, which further prove the unique benefits of our spectral-spatial acceleration methods. As seen in the Z-spectra in revised **Fig.6c**, only the CEST frequency range (1-3ppm) changes but not the water line-width or the other MTC range. The unique full-spectral scan of SSE-CEST allow us to retrospectively analyze the data, i.e., plot the spectral profile at each time-point and choose the spectral peak offsets (2.1ppm here) accordingly. This offset also adopted for the 2 μ T SSE-CEST protocol when compared with FDG-PET.

Figure 6. SSE-CEST for dynamic glucose imaging during oral glucose tolerance test (OGTT) experiments. a, MRI scan protocol of OGTT experiments, which consists of 13 SSE-CEST scans and 4 blocks of anatomical scans including a 3D T₁ weighted scan, a 2D T₁ mapping scan, a 2D T₂ mapping scan, and a 2D B₁ mapping scan. **b**, Comparison for liver Z-spectra and MTR_{asym} spectra between Baseline and 51-min post-OGT, indicating the peak

at 2.1ppm. **c**, Representative dynamic glucose-weighted CEST images and the corresponding time course, for glucose CEST changes (MTR_{asym} 2.1ppm, all time points subtracted the initial value before glucose take-up, error bar: value standard deviation among all motion-stabilized liver voxels). **d**, same layout as **c**, but for the corresponding T2 maps of the slice shown in **c**. The time-courses for other subjects are displayed in **Supplement Fig.6.1** and all-time-point images of the same subject are displayed in **Supplement Fig.6.2.**, depicting the location of pancreas.

Supplementary Figure 6.1. Dynamic changes in glucose CEST signal and T2 values during oral glucose tolerance (OGT) experiments. Data are shown for all five volunteers. **a**, Change in glucose-weighted CEST contrast, quantified as **MTR asymmetry at 2.1 ppm** relative to the pre-glucose baseline (Δ MTR_{asym}). All subjects demonstrate a clear rising trend following glucose ingestion. **b**, Corresponding T2 values plotted over the same time course. No consistent or significant change in T2 is observed, confirming that the dynamic Δ MTR_{asym} signal reflects specific metabolic changes rather than nonspecific variations in tissue water relaxation. The curves are presented as mean \pm std.

Regarding to the rationale of applying SSE-CEST in OGTT experiments, **we designed this experiment in order to expand the applications of our fast imaging in detecting the kinetics of exogenous CEST contrast agents.** Indeed, the oral glucose tolerance test (OGTT) experiments may not require a rapid acquisition technique in 10-20 seconds time scale. But the main benefit of this single-breath-hold abdominal imaging was the motion-stabilizing (motion-frozen) effect. From the search results from Web of science, dynamic glucose enhancement experiments are rising each year but only 1 abdominal study on due to the technical barrier (**Knutsson, Xu et al. 2023**). In the published human brain tumor studies intra-venously injection of glucose was used with a much-faster and clean kinetic course, where our methods could be applied. But for the very 1st technical paper, it's difficult to apply for the IRB approval using vessel injection of glucose. Additionally, the SSE-CEST could be applied in other exogenous agents including Iodine-based CT agents for tumor pH imaging(**Anemone, Consolino et al. 2021**).

As seen in **Supplementary Fig. 6.2**, we got nice and well-aligned images for each time point post OGTT. Besides, we also displayed the multi-organ images in the Supplement. With the pancreas clearly depicted, which may be useful for diagnosis of pancreas diseases(**Jardim-Perassi, Irrera et al. 2023**).

Supplementary Figure 6.2. The ST (2.1ppm) images acquired at all time-points, for the same subject the same slice as in Fig.6. Images displayed well-alignment, clearly depicting the pancreas (white arrow).

Author Actions:

1. Removed the previous Line-width plots, which was improper for using 2uT data.
2. Improved the SNR of MTRasym(2ppm) plot by applying a quality-control procedure, by removing the oscillating voxels which may be affected by cardiac or motions other than breathing. Replotted the OGTT figures and time courses, with the error bars only including the liver voxels post above filtering (revised Fig.6c, Supplementary Figure 6.1).
3. Add the spectral-comparison acquired pre- and 51 minutes post-OGT, which clearly indicates the maximum changes occurring at 2.1ppm (MTRasym used for glucose CEST, revised Fig.6b)
4. Add **Supplementary Fig. 6.2** of all time-point images, indicating the good spatial reconstruction with well-alignment in between. The depiction of multi-organs including pancreas reveals future applications in other application scenarios upon i.v. administration of exogenous agents (Discussion 4th paragraph). Discussion also has strengthened that “the unique full-spectral scan of SSE-CEST allows us to retrospectively analyze the data, i.e. plot the spectral profile at each time-point and choose the spectral peak offsets (2.1ppm here) accordingly. This offset is also adopted for the 2uT SSE-CEST protocol when compared with FDG-PET.”

[Comment 1.10]

10. Similar to Figure 5, there is a significant difference in the ST(3.5ppm) results but no significance in the APTw. Are the results due to changes in water parameters or MT instead of a CEST effect?

Author Response: This comment is closely related to Comment 1.8, please see our detailed response and actions there.

[Comment 1.11]

11. In some figures, it is not clear how many adjacent slices/points were used in the SVD. Using adjacent slices in SVD requires anatomical similarity. For heterogeneous regions (e.g., liver lesions) how would this affect the results?

Author Response: Also see R2.1, which displayed comparison between patient images reconstructed using single slice, 3, 5, and 7 adjacent slices. **We have now clarified in the method. i.e., For phantom**

SVD-based reconstruction utilized 5 adjacent slices with 21 slices acquired in total. In patients we utilized 5 adjacent slices with 41 slices in total (reconstructed from 21 acquisition steps post zero-filling). We assume it does not affect the anatomical images nor the spectral shapes according to the partial-separable theorem. **Besides, we do not perform any truncations that would discard components representing detailed spectral features or anatomical details.** i.e., The main purpose of putting 5 adjacent slices are frequency-augmentation for the 8-10 acquired z - ω lines within single breath-hold, i.e., achieving 5X sampled ω to further improve the accuracy of spectral reconstruction/interpolation.

Supplementary Figure 1.3. Influence of adjacent-slice number on reconstruction accuracy in ex vivo porcine liver. **a**, Reconstruction error plotted across frequency offsets (ω) when different numbers of adjacent slices (N_{adj}) are included in the spectral-basis interpolation, which is quantified as the voxel-wise absolute difference between the reconstructed data and the conventional CEST acquisition (gold-standard). **b**, Mean reconstruction error as a function of the total number of adjacent slices used in the interpolation. The error decreases as more adjacent slices are incorporated and plateaus when the adjacent slice number reaches five—the configuration adopted for all reconstructions in this study—indicating that the interpolation is robust and that additional slices provide diminishing improvements. Note that the number of adjacent slices includes the target slice itself.

Briefly, our low-rank reconstruction assumed that the voxels within these adjacent slices exhibit similar Zspectra features and get enough voxel numbers for reliably extracting the basis function of Zspectra. Therefore, the SSE-CEST reconstruction presumably does not have strict requirement of anatomical similarity of adjacent slices and still maintained fine structural details (i.e., the detailed vessel structure is displayed in Fig.4, Fig.R6(old Fig.5)). According to our data from tumor patients, it could well-reconstruct the small lesions. Below is an example of all slices for the patient in Fig.7.

Supplementary Figure 7.3 All 41 slices of ST (3.5ppm) acquired from SSE-CEST ($B_1 = 0.7 \mu\text{T}$) for an HCC patient post-treatment (The same patient as in Fig. 7). The white arrows denote the active lesions exhibited higher amide signals (hyper-intensity in orange color), whereas the green arrows denote the necrotic lesions post-treatment, showing a lower amide signal than normal control tissues (hypo-intense in blue color). The lesions matched well with the DCE images (revised Fig. 7, bottom row).

Supplementary Figure 7.4. The represented Z spectra and LD spectra for active lesion region, and for the normal control region (The same patient as in Fig. 7)

Author Actions:

1. Added the ST(3.5ppm) images of all-slice from the same patient in Fig.7, which displayed well spatial-reconstruction with hyper-intensed foci indicating active tumors and hypo-intensed region suggesting the necrosis.
2. Figure 7a,b comes from the same patient with different slices, which displayed well lesion structure (heterogenous regions)
3. Added Supplementary Fig.7.3, with the Zspectra and LD spectra comparison between the denoted active tumor, necrosis, as well as the normal-appearing regions on the same slice.

[Other comments]

[Comment 1.12]

12. In the supplementary, can you please describe how the resolution is calculated in z-omega space?

Author Response: Also see R2.1. Thanks for the suggestions, we have added the resolution calculation in supplementary methods (part 3), with Zneg(0.7uT) and Zpos(0.7uT) human SSE-CEST trajectories also displayed in supplement 5.1. The resolution in z- ω space is intrinsically determined by the spectral-spatial encoding relationship $\Delta\omega = \gamma \cdot G_{sat} \cdot \Delta z$. Briefly, the slice thickness (Δz), namely the spatial resolution along the z-axis, is calculated as the slice-direction field of view (FOV_s) divided by the number of slices (41 reconstructed slice post zero-fitting with 21 acquired steps for single-breathhold liver imaging). For a fixed FOV and number of slices, the acquired spectral steps $\Delta\omega$ is

directly proportional to the smallest non-zero G_{sat} . **This coupling between spatial and spectral dimensions is a fundamental characteristic of the SSE-CEST encoding scheme.** For instance, in the in vivo parallel-line acquisition protocol, the spatial resolution along z (Δz) is the slice thickness, given by the FOV (150 mm) divided by the number of slices (41), resulting in $\Delta z = 3.66$ mm; The spectral resolution ($\Delta\omega$) is then determined by the smallest saturation gradient used ($G^{\text{sat}} = -0.1$ mT/m) and Δz . This yields a native spectral resolution of $\Delta\omega \approx 0.1$ ppm.

Author Actions:

1. Added a paragraph of description in Supplement methods part 3.

1. Added z-w labels in Fig 2 of porcine liver;

[Comment 1.13]

13. There is inconsistent use of delta omega (e.g., Figure 1d and Eq. 1)

Author Response: We sincerely thank the reviewer for catching this oversight. We have now carefully revised the manuscript to ensure the symbol for the saturation frequency offset from water, ω , is used consistently throughout the text, figures (including Figure 1d), and equations (including Eq. 1).

Author Actions: carefully checked throughout the entire manuscript, and corrected.

[Comment 1.14]

14. I assume the total scan time is calculated by the number of saturation encodings multiplied by the shot interval. Based on the methods, this will mean a scan time of approximately 20 seconds. Can the authors clarify this?

Author Response: We thank the reviewer for this careful calculation. Yes, the total acquisition time is indeed determined by number of saturation encodings \times shot interval. The 0.7uT in vivo protocols involved 10 z-w lines plus S0 (Supplementary Table1, Fig.5.1), while the 2uT protocol acquired 11 z-w lines plus S0. Previously when we calculated the minimum scan time, the time for $\omega = 0$ ppm and 1560 ppm (S0) were not included since the S0 could be registered from another scan and the $\omega = 0$ ppm was not necessary.

However, since in this paper, we acquired 12 lines for the 2uT and 11 lines for the 0.7uT, with most of subject tolerating the breath-hold time. Now we clarify the actual acquisition time involved (17-21 seconds), with the minimum scan time of "12-15 seconds".and i.e. 12 encodings \times 1472 ms = ~ 17.7 seconds for $B1 = 2 \mu\text{T}$ protocol, and 11 encodings \times 1871 ms = ~ 20.6 seconds for $B1 = 0.7 \mu\text{T}$.

We would like to clarify two key points regarding scan time flexibility:

- 1) **Parameter Adjustability:** The breath-hold duration is directly tunable. For instance, reducing the number of z- ω encodings in the 2 μT protocol from 12 to 10 would bring the scan time to 14.7 seconds, well under the 15-second threshold. The 0.7 μT protocol requires a longer shot interval to accommodate its longer saturation pulse, hence the slightly longer acquisition.
- 2) **Synergy with Other Accelerations:** As demonstrated in **Supplementary Figure 1.6**, combining SSE-CEST with a 50% keyhole acceleration reduced the number of required k-space lines per encoding by half. This successfully achieved a total scan time of (11 encodings \times 1472 ms / 2) +

1472 ms = 9.6 seconds for a full dataset, proving that sub-10-second 3D metabolic imaging is feasible with our framework.

The slightly longer times reported here represent a choice that prioritized spectral coverage and SNR for this initial comprehensive evaluation. The method inherently supports shorter acquisitions, as shown, to suit specific clinical needs. We have revised the manuscript to state the precise scan times and highlight this flexibility.

Author Actions:

Clarified the actual acquisition time involved (18-21 seconds), with the minimum scan time of "12-15 seconds".

Reviewer 2:

Overall comments: This is very interesting research, combining UFZ and low rank tensor-based reconstruction to achieve super-fast abdominal CEST imaging within a single breath-hold. The study was well-planned, including optimization of different z- ω encoding patterns, validation in glycogen phantoms, and in vivo studies in healthy subjects, fasting, glucose tolerance tests, and hepatic carcinoma patients (both HCC and necrotic lesions). Imaging framework is well presented, manuscript is very well prepared, and results are convincing. This technique has great potential of making CEST imaging feasible in the abdomen. I have a few questions:

[Comment 2.1]

1. The major technical innovation is z- ω encoding, allowing x11 acceleration. It relies on interpolation of the spectral basis from adjacent slices, assuming the CEST signals are similar. It could be ok for normal subjects or diffuse disease but may have problems when there are metabolite concentration variations or focal disease. Zcorr was solved directly from the interpolated spectral basis without further refinement. The reliability of this approach needs to be rigorously tested.

Author Response: We appreciate the reviewer's positive feedback. We also thank the insightful comment regarding to the core reconstruction assumption and its reliability in focal disease. We agree that this is a critical issue and should be rigorously addressed and tested. **We will respond in three aspects as below.**

(1) Rationale for spectral reconstruction using adjacent slices: inherent tolerance for heterogeneity under partially separable theory (Also see R1.11): The idea of SSE-CEST reconstruction is similar to that used in MR spectroscopic imaging, based on the subspace model termed as Partially Separable Functions (PSF) theory, a mathematical framework for representing high-dimensional spatiotemporal data in a low-dimensional subspace. Since the time scale of single-breath-hold only allows acquiring 8-12 lines, which means each slice only sampled 8-12 frequency offsets. SSE-CEST does not assume uniform spatial contribution of CEST signals across the entire FOV or across the adjacent slices. Instead, the main purpose of putting together adjacent slices is the augmentation for sampled frequencies, ensuring that each group of voxels got 3X sampled offsets for spectral details. Briefly, we NOT directly put the value of adjacent frequencies ($\omega \pm \Delta \omega$) to a certain voxels. But for SVD feature extraction, we combined voxels within three adjacent slices, forming a matrix of $M \times 3V$, where M denoted the normalized z-w lines and $V = N_x \times N_y$ denotes the in-slice voxel numbers. As long as the adjacent slices containing voxels exhibiting similar spectral features (belonged to the same group), their spectral and spatial details could be reconstructed with high-fidelity. Noted that **we do not perform any truncation that would discard components representing spectral or spatial details.** For example, for the in vivo 2uT protocol of human abdomen, CEST datasets from 5 reconstructed slices are put together for SVD, each with 11 distinct sampled offsets. (five sets of 11 offsets \times 6400 voxels, each set with slight-shifted distinct offsets). Therefore, the resulting spectral basis functions, therefore, collectively represent the full range of spectral shapes

within the grouped voxels, i.e. both normal tissue and any potential focal lesions present within the adjacent slices. In summary, SSE-CEST does not change the acquired resolution on Z-dimension, either for the in-plane resolution. The spectral resolution mainly determined by Eq.1, which is 0.1 ppm for in vivo case (sufficient for 3T CEST spectra). **We have now added description of SSE reconstruction and calculation for resolution in the Supplementary method 1.** Additionally, we also compared the reconstruction quantity using different numbers of adjacent slices (new **Supplementary Fig.7.5, same patient as in main Fig.7**). The results suggest that the images reconstructed using $N_{adj} = 3, 5, 7$ are almost identical, for all the four frequency offsets (next page)

Supplementary Figure 7.5 Comparison of SSE-CEST reconstruction using different num. of adjacent-slice. (same patient as in Fig.7)

(2) Reliability validations in phantom and in ex vivo porcine liver: Since in vivo scans lack a gold-standard without motion, we have enhanced the reliability test in ex vivo porcine liver and in the phantom experiments, by comparing with conventional CEST. We have reperformed the phantom experiment by dissolving various concentration of glycogen in 1% agar (previous in PBS, **See R1.5**). In the revised version, ST images from SSE-CEST and conventional CEST were comprehensively compared across all slices. The two methods exhibited visually identical contrast, with strong linear

correlations observed both between their signals and between SSE-CEST signals and glycogen concentrations. These results demonstrate the high-fidelity reconstruction capability of SSE-CEST (**Supplementary Fig. 3.1-3.3, revised Fig.3**). In ex vivo porcine liver, we also displayed all 21 slices from SSE-CEST, with all three ST maps revealing well anatomical reconstruction of vessel details, and good consistency with the conventional CEST (**Supplementary Fig.2.3, revised Fig.2**). **As seen from Supplementary Fig.3.4, adjacent slices sampled distinct frequencies but exhibit almost-identical Z-spectra.** In porcine liver, we also displayed the spectral errors and the mean errors using different number of adjacent-slice (1-7), with the conventional CEST as gold standard (**Supplementary Fig. 1.3**).

(3) Reconstruction of Spatial details in healthy volunteers and in patients with focal liver disease. As described in part 1 theory, SSE-CEST does not change the readout spatial resolution. While the spectral bases are shared, the spatial coefficient map (the U matrix in Eq. 7) determines how these bases are weighted to reconstruct each voxel. The reconstruction algorithm independently solves for the coefficients of each voxel, meaning that the presence of a focal lesion does not corrupt the spectral reconstruction of surrounding normal tissue, and vice-versa. As seen on the ST images acquired under both B1's (Supplementary 5.6), the anatomical details of major vessels are clearly observed, and the multi-organ could be easily recognized and overlapped on the high-resolution T1 images (Supplementary 5.7). For patients, SSE-CEST not just well localized the lesion as compared with the Gd-enhanced DCE images, but the intensity allowing differentiation active tumor recurrence from necrosis post-treatment (**Fig.6**). In Supplementary Fig. 6.1, we displayed all 41 reconstructed slices for the patient in Fig.6, (ST(3.5ppm), 0.7uT) with the metabolic-active lesions and the necrosis easily recognized (pointed by arrows of two different colors). Besides, the full-spectral acquisition capability allows clear delineation of the Zspectral shapes and LD spectral shapes, improving the interpretation and the quality control of CEST in vivo, especially in the abdomen. Interestingly, Zspectral analysis revealed at active malignant lesions exhibited a stronger MTC compared with the normal-appearing control regions in the same slice (**Supplementary R6.2**).

(4) Clinical validation with radiologist assessment: To quantitatively evaluate the clinical reliability of our reconstruction for focal disease, we analyzed data from 22 patients with two trained radiologists independently contouring lesions directly on the SSE-CEST ST maps while blinded to other imaging results. This analysis revealed excellent inter-reader agreement (Cohen's kappa = 0.89) and demonstrated clinically meaningful detection rates that logically correlated with expected metabolic activity: active HCC lesions (n=11) showed high detection rates of 100% and 90.9%; treated necrotic lesions (n=12) showed high detection rates of 100% and 100%; while benign lesions including cysts and hemangiomas (n=22) exhibited expectedly lower rates of 45.5% and 50%. This clear detection gradient—from high in metabolically active malignancies to low in benign entities—validates that SSE-CEST contrast reflects underlying tissue metabolism and confirms the robustness of our spectral interpolation approach in clinical scenarios with heterogeneous pathologies.

Supplementary Figure 7.2. All 41 slices of APT_w images using SSE-CEST 2 μT protocol (the same patient in Fig.7)

(5) Framework flexibility for improved specificity: We agree that the current in-plane resolution ($4.5 \times 4.5 \text{ mm}^2$) could limit the detection of very small lesions. While the low-resolution is a common bottle-neck in metabolic imaging, our SSE-CEST framework allows higher resolution. To demonstrate this, we performed an additional scan on a patient with a small liver cyst (**Supplementary Fig.7.6**). By reducing the FOV while maintaining the breath-hold time, we achieved an in-plane resolution of $3.5 \times 3.5 \text{ mm}^2$. The small lesion is clearly visible in the high-resolution SSE-CEST map, confirming that the framework can be optimized for specific clinical questions requiring higher spatial resolution.

Supplementary Figure 7.6 Demonstration of high-resolution SSE-CEST for depicting small lesions.

Additionally, we also added a new **Fig.8 (next page)**, by comparing SSE-CEST images with FDG-PET, a non-MR modality. The hyper-intense region on SSE-CEST amide maps and glucose maps displayed consistency with the hyper-metabolic PET hot spots.

In summary, while the reconstruction leverages spectral similarity, it is robust to anatomical and metabolic heterogeneity. SSE-CEST displayed high-fidelity spatial-spectral reconstruction in both phantom and ex vivo porcine liver experiments, as suggested by the comprehensive comparison with conventional CEST as gold standard. In tumor patients, it reveals well co-localization with Gd-enhanced DCE MR images (**Fig.7, and Supplementary Fig.7.2-7.4**), allowing differentiating active tumor recurrence from necrosis post-treatment. Combined with our clinical validation and the demonstrated flexibility to enhance resolutions, **we are confident in the reliability of the SSE-CEST approach for evaluating focal liver diseases.**

Author Actions:

- 1) Clarified reconstruction methods (**in results part 1 and in methods**) and added detailed calculation of spatial-spectral resolution. (**supplementary methods Part 1**).
- 2) Added all-slice images and comprehensive comparison with the conventional CEST in phantoms and in ex vivo porcine liver. (**supplementary fig.s 2.2,2.3, 3.1-3.5**)
- 3) Added new Fig. 8 with comparison with a non-MR modality, FDG-PET.(See **R1.1**)
- 4) Add an additional patient with two spatial resolution acquired, suggesting the capability of imaging

small lesions. Supplied all-slice APTw maps and the ST(3.5ppm) images under 0.7uT, demonstrating the reliable reconstruction and ability for focal lesions. (Supplementary Fig.7.1,7.2)

New Fig. 8 Comparison of SSE-CEST metabolic maps and FDG-PET in a patient with hepatocellular carcinoma (HCC), scanned post treatment. Three axial slices are shown, highlighting active HCC lesion (red arrows) and adjacent post-treatment necrotic regions (white arrows). **a**, anatomical T1 weighted image. **b**, SSE-CEST metabolic maps: (left) amide CEST (LD (3.5 ppm)) using B1 of 0.7 μ T, reflecting mobile protein/peptide content; (right) the ST image at 2.1 ppm (B1 of 2 μ T), reflecting glucose-related metabolism. **c**, Corresponding slices on the fused images of 18 F-DG-PET and CT. The hyper-intense regions on SSE-CEST images matched well with the 'hot-spot' regions on PET images, which indicates higher glucose metabolism.

[Comment 2.2]

2. Are saturation times of 0.4 and 0.8s sufficient for the CEST signal to reach steady state? Usually 2-4s are used for continuous saturation.

Author Response: We thank the reviewer for this important question. Indeed saturation time (T_{sat}) is an important acquisition parameter that is closely related to the reliability and the amplifying efficiency of CEST-MRI. In the SSE acquisition order, we took into account the limited T_{sat} by acquiring S_0 and edge frequencies at the beginning, ensuring that the spins reached steady-state later when 'labeling' the target frequencies. Although Zspectra for some voxels displayed rising tails at edge frequencies when acquired early (see **R1.1** and **Supplementary Figs. 5.2 and 5.3** for 0.7 μ T Z-spectra), these non-steady-state signals were corrected in the Lorentzian fitting stage for 0.7 μ T. For 2 μ T, we did not

observe such abnormal shapes in the Zspectra.

To further address the T_{sat} -dependence of SSE-CEST signals, we have performed simulations of SSE-CEST signals, using T_1 and T_2 within the relevant ranges of liver parenchyma (~800 ms as measured under 3T, **response R1.8**). In details, our simulations utilized Bloch-McConnell model including water pool, a slower-exchanging pool (amide, $\omega=3.5$ ppm, $k_{sw}=39$ Hz), and three faster-exchanging hydroxyl pools ($\omega=1.1, 2.1, 2.9$ ppm, $k_{sw}=1000$ Hz) five pools for water, amide, and three hydroxyl groups. Noted that the MT and the relayed NOE effects from macro-molecules were not considered as it does not affect the ‘labeling’ and the resulted signals at positive offsets of CEST (downfield of water). **Supplementary Fig.1.7 suggested that the sub-second saturation obtained a sufficient signal, i.e., > 80% of the values using $T_{sat}=2000$ ms.** The slice-by-slice comparison of ST images from SSE-CEST and from the conventional also exhibit identical images (**Supplementary Fig.3.2,3.3**).

Supplementary Fig.1.7 Numerical simulation of CEST signal evolution for fast- and slow-exchange metabolites under SSE-CEST conditions. The Bloch-McConnell simulation includes five pools for water, amide, and three hydroxyl groups. (a) M_{sat}/M_0 signal evolution at 2.1 ppm (representing fast-exchange pools such as hydroxyls) and (c) at 3.5 ppm (representing the slower-exchange amide pool) as a function of saturation time (T_{sat}). The initial magnetization started from the readout steady-state ($M_i=0.272$), reflecting the zero-recovery-time design of the SSE-CEST sequence, rather than from thermal equilibrium ($M_0=1$). (b, d) The corresponding MTRAsym dynamics at 2.0 ppm and 3.5 ppm, respectively. At the chosen in vivo T_{sat} , the transient $R_{2\rho}$ oscillations are fully damped, and the achieved MTRAsym represents a substantial fraction of the 2s-reference signal: 73.9% ($2 \mu T$) and 57.2% ($0.7 \mu T$) at 2.0 ppm, and 41.6% ($2 \mu T$) and 51.2% ($0.7 \mu T$) at 3.5 ppm.

Since SSE-CEST utilized a small-flip-angle TFE readout, with zero recovery-time among z-w lines. (TR per shot: 0.4s (T_{sat}) + 1.07s readout for $2\mu T$; T_{sat} of 0.8s plus 1.07s readout for $0.7 \mu T$) **Supplementary Fig.1.7** suggests the initial magnetization of SSE-CEST readout started from $M_i=0.272$, with respect to the thermal equilibrium ($M_0=1$). **This could also explain the low signals around +/-1.2 ppm in $2 \mu T$ Z spectra, given the over-estimated S_0 (See R1.6).** Nevertheless, given

the large water load for body imaging, the SNR is not an issue (as seen in **Supplementary Fig.5.4**, the post-meal images displayed higher SNR than the corresponding ones acquired post-fasting at all five offsets under both B1's).

However, we acknowledge that shorter Tsat may limit the ultimate specificity for certain applications (See R2.3 below). **The SSE-CEST framework offers inherent flexibility to address this.** For instance, the saturation time can be extended by reducing the number of z- ω encodings, a strategy we already employed in our 0.7 μ T protocol (Tsat = 0.8 s with 10 encodings). While existing free-breathing or multi-breath-hold techniques can achieve long Tsat and dense spectral sampling, they incur the costs of prolonged scan time, image misregistration, and lower clinical success rates—the very challenges our single breath-hold method is designed to overcome.

Additionally, we also added a QC pre-correction and extrapolated MTC according to the priors (Also see **R1.1**).

Author Actions:

(1) Added **Supplementary Figure 1.7** and the employed Dixon-TFE sequence **Supplementary Fig.4.1**.

[Comment 2.3]

3. There was discrepancy between phantom and in vivo studies. In human studies, B1 = 2 uT with 0.4 s saturation and B1 = 0.7 uT with 0.8 s saturation, with TFE shot intervals of 1472/1871 ms. By contrast, in phantom studies, 0.7 uT (3.5 s), 1 uT (3.5 s), and 2 uT (1.75 s) were used with a TFE shot interval of 8 s. It would have been helpful if they had included phantom experiments using the same parameters as in vivo, and compared them with a conventional CEST spectrum.

Author Response: Also see **R1.4**. We agree with the reviewer that matching phantom and in vivo parameters is crucial for validation. Following this suggestion, we have re-performed phantom studies. To better approximate the in vivo environment, the glycogen solutions were embedded in a 1% agar gel doped with Magnevist to adjust T1/T2 relaxation times.

Supplementary Figure 3.5 SSE-CEST results of glycogen phantom using human acquisition protocol. a, comparison of SSE-CEST Z-spectra with those acquired by conventional CEST b, phantom layout in x-y slices. c, representative glycogen NOE map of an axial slice from SSE-CEST. d, concentration dependence of glycogen NOE from SSE-CEST (n=11 slices, mean \pm std). All data were acquired with $B_1 = 0.7 \mu\text{T}$ and quantified using LD. Representative parameters: conventional CEST: $T_{\text{sat}}/TR = 3/10$ s; SSE-CEST: $T_{\text{sat}}/TR = 0.8/1.87$ s.

Author Actions:

1. **Added** phantom experiments using the same parameters as in vivo. (**Supplementary Figure 3.5**)
2. Discussed the results in comparison with the conventional CEST spectrum with a longer TR and delay time.

Reference:

- Anemone, A., L. Consolino, L. Conti, P. Irrera, M. Y. Hsu, D. Villano, W. Dastru, P. E. Porporato, F. Cavallo and D. L. Longo (2021). "Tumour acidosis evaluated in vivo by MRI-CEST pH imaging reveals breast cancer metastatic potential." *British Journal of Cancer* **124**(1): 207-216.
- Bie, C. X., S. W. Bo, N. N. Yadav, P. C. M. van Zijl, T. Wang, L. Chen, J. D. Xu, C. Zou, H. R. Zheng and Y. Zhou (2025). "Simultaneous monitoring of glycogen, creatine, and phosphocreatine in type II glycogen storage disease using saturation transfer MRI." *Magnetic Resonance in Medicine* **93**(4): 1782-1792.
- Chen, L., P. B. Barker, R. G. Weiss, P. C. M. van Zijl and J. D. Xu (2019). "Creatine and phosphocreatine mapping of mouse skeletal muscle by a polynomial and Lorentzian line-shape fitting CEST method." *Magnetic Resonance in Medicine* **81**(1): 69-78.
- Deshmane, A., M. Zaiss, T. Lindig, K. Herz, M. Schuppert, C. Gandhi, B. Bender, U. Ernemann and K. Scheffler (2019). "3D gradient echo snapshot CEST MRI with low power saturation for human studies at 3T." *Magnetic Resonance in Medicine* **81**(4): 2412-2423.
- Desmond, K. L., F. Moosvi and G. J. Stanisz (2014). "Mapping of Amide, Amine, and Aliphatic Peaks in the CEST Spectra of Murine Xenografts at 7 T." *Magnetic Resonance in Medicine* **71**(5): 1841-1853.

Dula, A. N., L. R. Arlinghaus, R. D. Dortch, B. E. Dewey, J. G. Whisenant, G. D. Ayers, T. E. Yankeelov and S. A. Smith (2013). "Amide proton transfer imaging of the breast at 3 T: Establishing reproducibility and possible feasibility assessing chemotherapy response." Magnetic Resonance in Medicine **70**(1): 216-224.

Glang, F., A. Deshmane, S. Prokudin, F. Martin, K. Herz, T. Lindig, B. Bender, K. Scheffler and M. Zaiss (2020). "DeepCEST 3T: Robust MRI parameter determination and uncertainty quantification with neural networks-application to CEST imaging of the human brain at 3T." Magnetic Resonance in Medicine **84**(1): 450-466.

Jardim-Perassi, B. V., P. Irrera, J. Y. C. Lau, M. Budzevich, C. J. Whelan, D. Abrahams, E. Ruiz, A. Ibrahim-Hashim, S. D. Erturk, D. L. Longo, S. A. Pilon-Thomas and R. J. Gillies (2023). "Intraperitoneal Delivery of Iopamidol to Assess Extracellular pH of Orthotopic Pancreatic Tumor Model by CEST-MRI." Contrast Media & Molecular Imaging **2023**.

Jin, T., P. Wang, X. P. Zong and S. G. Kim (2012). "Magnetic resonance imaging of the Amine-Proton EXchange (APEX) dependent contrast." Neuroimage **59**(2): 1218-1227.

Jones, C. K., D. Polders, J. Hua, H. Zhu, H. J. Hoogduin, J. Y. Zhou, P. Luijten and P. C. M. van Zijl (2012). "In vivo three-dimensional whole-brain pulsed steady-state chemical exchange saturation transfer at 7 T." Magnetic Resonance in Medicine **67**(6): 1579-1589.

Knutsson, L., X. Xu, P. C. M. van Zijl and K. W. Y. Chan (2023). "Imaging of sugar-based contrast agents using their hydroxyl proton exchange properties." Nmr in Biomedicine **36**(6).

Shizhen Chen, M. J., Yaping Yuan, Baolong Wang, Yu Li, Lei Zhang, Zhong-Xing Jiang, Chaohui Ye, Xin Zhou (2023). "Using endogenous glycogen as relaxation agent for imaging liver metabolism by MRI." Fundamental Research **3**(4): 481-487.

Xu, X., R. Leforestier, D. Xia, K. T. Block and L. Feng (2025). "MRI of GlycoNOE in the human liver using GraspNOE-Dixon." Magnetic Resonance in Medicine **93**(2): 507-518.

Yadav, N. N., J. D. Xu, A. Bar-Shir, Q. Qin, K. W. Y. Chan, K. Grgac, W. B. Li, M. T. McMahon and P. C. M. van Zijl (2014). "Natural D-Glucose as a Biodegradable MRI Relaxation Agent." Magnetic Resonance in Medicine **72**(3): 823-828.

Zaiss, M. and P. Bachert (2013). "Chemical exchange saturation transfer (CEST) and MR Z-spectroscopy : a review of theoretical approaches and methods." Physics in Medicine and Biology **58**(22): R221-R269.

Zaiss, M., B. Schmitt and P. Bachert (2011). "Quantitative separation of CEST effect from magnetization transfer and spillover effects by Lorentzian-line-fit analysis of z-spectra." Journal of Magnetic Resonance **211**(2): 149-155.

Zhou, Y., P. C. M. van Zijl, X. Xu, J. D. Xu, Y. G. Li, L. Chen and N. N. Yadav (2020). "Magnetic resonance imaging of glycogen using its magnetic coupling with water." Proceedings of the National Academy of Sciences of the United States of America **117**(6): 3144-3149.

Single-breath-hold 3D abdominal metabolic MRI: label-free imaging for enhanced liver disease diagnosis

(Manuscript ID: NCOMMS-25-48838-A)

We sincerely thank the Editors and Reviewers for accepting our paper **in principle**. We have revised the manuscript to address all editorial requests (detailed in the attached Author Checklist), with **the remaining comment from reviewer 1** responded below.

Reviewer 1

The authors have done a commendable job addressing my previous comments. I feel that the changes make the research much stronger and it is suitable for publication.

a minor edit: The broad direct water saturation line is not Lorentzian in nature and thus is probably the major reason for the poor fits in Supplementary Figures 5.1-5.3. Perhaps the authors can briefly mention in the Discussion or Section 5 of the supplementary that Gaussian lineshapes or multiple lineshape fitting may improve the specificity of the signal extracted.

Author Response:

We agree that refined quantification strategies should be discussed. Two edits have been made as below.

We have added in the Discussion that “Finally, our single Lorentzian approximation for water saturation at 0.7 μ T overlooks line broadening and magnetization transfer effects that become prevalent at higher B_1 amplitudes. Extension to Gaussian, Voigt, or multi-pool lineshapes is warranted to ensure fidelity across diverse saturation conditions and tissue types.” (**The last two sentences in the limitation part**)

Furthermore, we added descriptions in the supplement methods (the last paragraph in Session 2): “Noted this model neglects water line broadening and the distortions due to magnetization transfer contributions, therefore may cause inaccurate quantification for diverse tissue types and/or larger saturation B_1 field. Nevertheless, owing to the highly-efficient spectral interpolation and motion-stabilizing of SSE-CEST, other refined fitting strategies could be integrated in future— including Gaussian, Voigt, or multi-component Lorentzian schemes — combined with machine-learning (Chen, 2023) or deep-learning (Chen, L 2020, Glang, Deshmane et al. 2020).”

Chen, L., Schär, M., Chan, K.W.Y. et al. In vivo imaging of phosphocreatine with artificial neural networks. Nat Commun 11, 1072 (2020). <https://doi.org/10.1038/s41467-020-14874-0>

Chen Y, Dang X, Zhao B, Chen Z, Zhao Y, Zhao F, Zheng Z, He X, Peng J, Song X. Frequency importance analysis for chemical exchange saturation transfer magnetic resonance imaging using permuted random forest. NMR Biomed. 2023 Jun;36(6):e4744. doi: 10.1002/nbm.4744.

Glang, F., A. Deshmane, S. Prokudin, F. Martin, K. Herz, T. Lindig, B. Bender, K. Scheffler and M. Zaiss (2020). "DeepCEST 3T: Robust MRI parameter determination and uncertainty quantification with neural networks-application to CEST imaging of the human brain at 3T." Magnetic Resonance in Medicine **84**(1): 450-466.